# Mining Useful General Data for Low-Resource Domain Adaptation

Pingjie Wang [* 1]   Hongcheng Liu [* 1]   Yusheng Liao [1]   Ziqing Fan [1]   Yaxin Du [1]   Shuo Tang [1]
Yanfeng Wang [1]   Yu Wang [1]

Official repository: `https://github.com/applewpj/NTK-Selector`

## Abstract

Adapting large language models (LLMs) to low-resource domains remains challenging due to the scarcity of domain-specific data. While in-domain data is limited, vast general-domain data shares similar question-answer formats and reasoning patterns with domain tasks. This observation raises an important question: ***can useful general-domain data be mined to improve low-resource domain adaptation?*** Our initial findings show that general-domain chain-of-thought data contains useful auxiliary signals for domain adaptation, even without careful selection, motivating a new paradigm beyond exclusive reliance on domain-specific data. To systematically identify the most beneficial general-domain samples, we propose NTK-Selector, motivated by the Neural Tangent Kernel's ability to capture alignment in training dynamics. Since directly applying NTK to pretrained LLMs is impractical, we introduce a Jacobian-free NTK approximation and empirically demonstrate stable NTK-like behavior during fine-tuning. Extensive experiments across medical, financial, legal, and psychological domains demonstrate that NTK-Selector consistently outperforms domain-only fine-tuning and existing data selection baselines. In particular, NTK-Selector achieves gains of **+8.7** and **+5.1** points on Llama3-8B-Instruct and Qwen3-8B, respectively, compared to only **+0.8** and **+0.9** points from domain-only fine-tuning.

## 1. Introduction

The emergence of large language models (LLMs) has led to remarkable advancements across a wide spectrum of natural

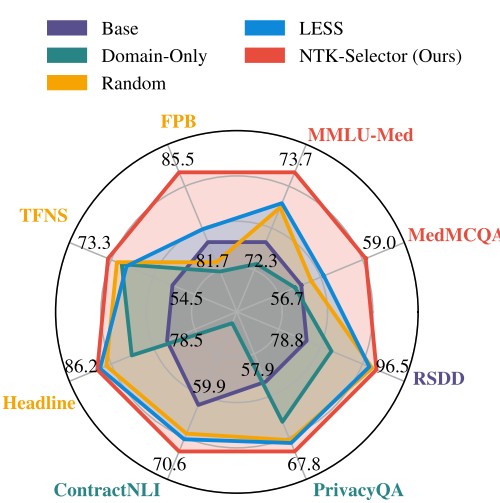

*Figure 1.* Performance of LLAMA3-8B-INSTRUCT evaluated on medical, financial, legal, and psychological tasks. Each task is augmented with 9K auxiliary samples selected by Random, LESS, and NTK-Selector from Cot Collection based on 1K domain points.

language processing tasks (Touvron et al., 2023; Chowdhery et al., 2023; Yang et al., 2025). However, their formidable capabilities are predominantly anchored in the availability of immense, high-quality pre-training and instruction-tuning datasets. This dependence is particularly problematic in low-resource domains, where data is scarce, expensive to curate, and often entangled with privacy constraints. In such settings, directly fine-tuning LLMs on limited domain-specific datasets frequently induces severe overfitting and poor generalization (Liu et al., 2023a; Béthune et al., 2025), as reflected by the degradation in Figure 1. Consequently, the utility of LLMs for specialized downstream applications remains fundamentally constrained.

While in-domain data is limited, there exists a vast amount of general-domain data that shares similar topics, question–answer formats, and reasoning patterns with domain tasks. This observation raises a natural and important question: ***can useful general-domain data be mined to improve low-resource domain adaptation?*** To explore this question, we conduct preliminary experiments using randomly sampled general-domain auxiliary data. As shown in Figure 1,

[*]Equal contribution [1]School of Artificial Intelligence, Shanghai Jiao Tong University, Shanghai, China. Correspondence to: Yu Wang <yuwangsjtu@sjtu.edu.cn>.

*Proceedings of the 43rd International Conference on Machine Learning*, Seoul, South Korea. PMLR 306, 2026. Copyright 2026 by the author(s).

even naive incorporation of such data yields consistent improvements over domain-only fine-tuning. This result provides an important insight: general-domain data contains latent auxiliary signals that can enhance domain performance, motivating a new paradigm for domain adaptation beyond exclusive reliance on domain-specific supervision.

In practice, exploiting general-domain data requires principled data selection. Existing methods (Xie et al., 2023; Liu et al., 2024), such as LESS (Xia et al., 2024), are designed for task-specific settings and rely on large in-domain data pools or curated validation sets, assumptions that do not hold in low-resource, cross-domain scenarios. As a result, they are not directly applicable to mining useful general-domain data for domain adaptation.

To bridge this gap, we seek a criterion to predict how training on a general-purpose instance will influence the target performance. The Neural Tangent Kernel (NTK) (Jacot et al., 2018b) provides a principled signal for this purpose, as it characterizes the alignment of training dynamics induced by different samples, independent of explicit semantic similarity. Under standard NTK assumptions, training dynamics are governed by the kernel at initialization, allowing each sample's influence to be estimated once without recomputation or enumeration over combinations during training.

Applying NTK-based methods to modern pretrained LLMs, however, is non-trivial. Classical NTK formulations rely on assumptions that are not directly satisfied in pretrained, finite-width models, and exact NTK computation is computationally prohibitive at scale. In this work, we show that these challenges can be effectively addressed by empirically demonstrating stable NTK-like behavior in LLMs during fine-tuning and introducing a Jacobian-free NTK approximation that makes NTK-based selection practical for LLMs, which go beyond Mohamadi & Sutherland (2022).

Based on these key insights, we propose **NTK-Selector**, a novel two-stage data selection framework. It first performs a coarse-grained pre-selection of candidates using embedding similarity. This is followed by a fine-grained NTK selection stage, where we compute the NTK utility score for each pre-selected sample. Our method incorporates LoRA-based training and random gradient projections to significantly reduce memory and computational overhead, enabling efficient selection of large-scale auxiliary data. To our knowledge, this is the first work to both empirically investigate the kernel behavior of LLMs in the fine-tuning regime and to leverage the insight for scalable data selection. In summary, our contributions are summarized as follows:

- We identify and formalize a new domain adaptation paradigm that mines useful auxiliary data from large-scale general-domain corpora to improve performance in low-resource domains, under the realistic constraint of having no task-specific validation sets or large in-domain data pools available (§2.1).

- We propose **NTK-Selector**, a principled and efficient two-stage data selection framework grounded in the Neural Tangent Kernel. To enable NTK-based selection for pretrained LLMs, we provide the first empirical evidence of stable NTK-like behavior during fine-tuning and introduce a Jacobian-free NTK approximation that makes NTK computation practical at scale, going beyond prior NTK approximations designed for idealized settings (§3, §4).

- We demonstrate that NTK-Selector delivers substantial and consistent performance gains across diverse low-resource domains, including medicine, finance, law, and psychology. In particular, it achieves up to **+8.7** and **+5.1** average improvement over domain-only fine-tuning on LLAMA3-8B-INSTRUCT and QWEN3-8B, respectively (§5, §6).

## 2. Preliminaries

### 2.1. Problem Statement

Formally, let $\mathcal{D}$ denote the small domain-specific dataset and $\mathcal{C}$ the large general-purpose candidate corpus, where $|\mathcal{D}| \ll |\mathcal{C}|$. We aim to select a subset $\mathcal{S} \subset \mathcal{C}$ of size $N$. Let $f(\mathbf{x}, \theta) : \mathbb{R}^d \to \mathbb{R}^D$ represent the output of LLM parameterized by $\theta$ with input $\mathbf{x} \in \mathbb{R}^d$, $\mathcal{L}(\theta; \mathcal{X})$ denote the training loss on a dataset $\mathcal{X}$, and $T(f(\cdot; \theta), \mathcal{T}_{\text{test}})$ denote the task performance metric evaluated on a held-out test set $\mathcal{T}_{\text{val}}$. Our objective is to identify an optimal subset $\mathcal{S}^*$ that maximizes downstream task performance after fine-tuning on the combined dataset $\mathcal{D} \cup \mathcal{S}^*$. This can be formulated as the following bi-level objective function:

$$\mathcal{S}^* = \arg \max_{\mathcal{S} \subseteq \mathcal{C}, |\mathcal{S}|=N} T(f(\cdot; \theta_{\mathcal{S}}), \mathcal{T}_{\text{test}}),$$
$$\text{where} \quad \theta_{\mathcal{S}} = \arg \min_{\theta} \mathcal{L}(\theta; \mathcal{D} \cup \mathcal{S}). \tag{1}$$

### 2.2. Neural Tangent Kernel

The Neural Tangent Kernel (NTK) (Jacot et al., 2018b) is a theoretical framework that proves that the evolution of an infinitely wide neural network under specific parameterization can be described by a linear kernel regression. This provides a powerful tool for analyzing the dynamics and generalization of deep neural networks. Formally, for a model $f(\cdot; \theta_t)$ with parameters $\theta_t$ in the training step $t$ and two input examples $\mathbf{x}, \mathbf{x}' \in \mathbb{R}^d$, the NTK $\Theta(\mathbf{x}, \mathbf{x}'; \theta_t)$ is defined as:

$$\Theta(\mathbf{x}, \mathbf{x}'; \theta_t) = \langle \nabla_{\theta_t} f(\mathbf{x}; \theta_t), \nabla_{\theta_t} f(\mathbf{x}'; \theta_t) \rangle, \tag{2}$$

where $\nabla$ denotes the gradient and $\langle \cdot \rangle$ represents inner product. A high NTK value indicates that $\mathbf{x}$ and $\mathbf{x}'$ will guide

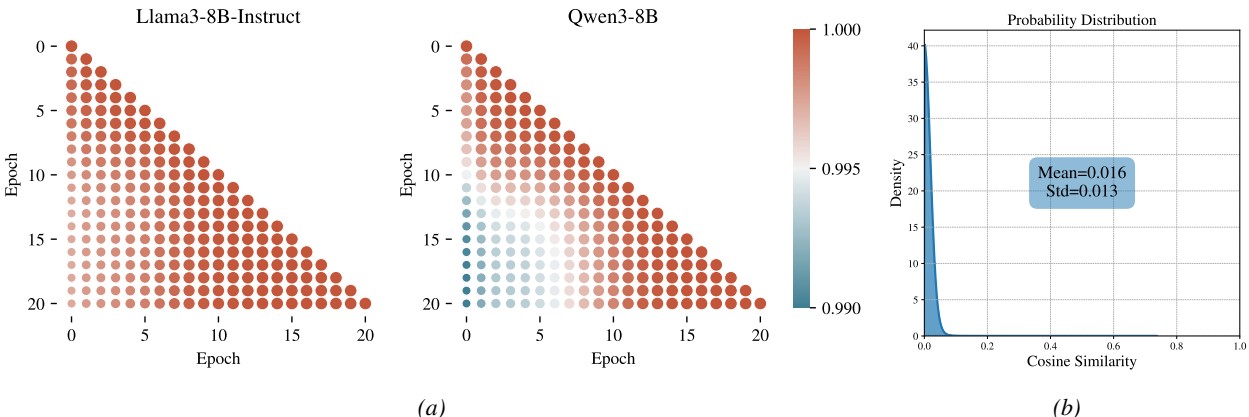

*Figure 2.* (a) Frobenius cosine similarity between NTK of LLAMA3-8B-INSTRUCT and QWEN3-8B during LoRA-based instruction tuning towards financial sentiment analysis task. (b) Gradient similarity distribution of cross output across 100 samples randomly sampled from ContractNLI task.

the model in similar directions during training, leading to correlated generalization behavior. This makes NTK an ideal metric for selecting the most relevant cross-domain auxiliary data. Under the infinite-width and Gaussian initialization regime, the NTK remains constant throughout training, so the kernel computed at initialization governs each sample's influence over the entire optimization trajectory. This invariance allows sample influence to be estimated once, without recomputation or enumeration of sample combinations, making NTK a scalable and principled criterion for cross-domain data selection.

## 3. NTK Applications for LLMs

While the NTK serves as a powerful tool for data selection, its direct application to LLMs faces two key challenges: (i) LLMs don't meet the theoretical assumptions of NTK, and (ii) the exact NTK computation is infeasible. In this section, we address these challenges by empirically demonstrating an NTK-like behavior in LLMs (§3.1) and proposing a scalable approximation method (§3.2).

### 3.1. NTK-like Behavior in LLMs

While classic NTK theory assumes a Gaussian initialized infinite-width network with a constant kernel matrix throughout training, real-world pre-trained LLMs operating in the fine-tuning regime do not satisfy these assumptions. Our empirical observations reveal that the NTK of a fine-tuning LLM is not strictly constant; however, its directional structure remains remarkably stable. This phenomenon is characterized by the time-evolved kernel being nearly collinear with the initial kernel, as quantified by a high Frobenius cosine similarity (e.g., greater than 0.99), which is visualized in Figure 2a. We formalize this observation as NTK-like behavior.

**Definition 3.1** (NTK-like). Given a model $f(\cdot; \theta_t)$ with NTK $\Theta(\cdot, \cdot; \theta_t)$, the model is said to exhibit NTK-like behavior on the interval $[0, T]$ if there exists a small constant $\epsilon \in (0, 1)$ such that for all $t \in [0, T]$, the Frobenius cosine similarity between the time-evolved NTK and the initial NTK satisfies:

$$\cos_F(\Theta(\cdot, \cdot; \theta_t), \Theta(\cdot, \cdot; \theta_0)) \geq 1 - \epsilon. \quad (3)$$

This NTK-like property suggests that during fine-tuning, the model primarily amplifies feature directions established at initialization rather than learning entirely new, orthogonal features. We formally show that under this condition, training is equivalent to optimization with a fixed kernel $\Theta(\cdot, \cdot; \theta_0)$ plus a controllable perturbation term. Theoretical results are presented in the following theorem with proofs in Appendix C. Empirical results on more model architectures, tasks, and finetuning strategies are detailed in Appendix H.1.

**Theorem 3.2.** *Given a model $f(\cdot; \theta_t)$ with NTK $\Theta(\cdot, \cdot; \theta_t)$ whose training dynamics are governed by a gradient flow $\dot{f}(\cdot; \theta_t) = -\eta\Theta(\cdot, \cdot; \theta_t)\gamma(t)$, where $\gamma(t) = \nabla_f \mathcal{L}(f(\cdot; \theta_t))$ and $\eta$ is the learning rate, if it exhibits NTK-like behavior on $[0, T]$, its training dynamics can be re-parameterized onto a new time axis $u$:*

$$\dot{f}(\cdot; \theta_{t(u)}) = -\eta\Theta(\cdot, \cdot; \theta_0)\gamma(t(u)) + \Delta(u), \quad (4)$$

*where $u(t) := \int_0^t a^*(\tau)d\tau$, $a^*(t) = \frac{\langle\Theta(\cdot, \cdot; \theta_t), \Theta(\cdot, \cdot; \theta_0)\rangle}{||\Theta(\cdot, \cdot; \theta_0)||^2}$, and the perturbation term $\Delta(u)$ is bounded with $||\Delta(u)|| \leq \eta||\Theta(\cdot, \cdot; \theta_0)||\frac{\sqrt{2\epsilon}}{1-\epsilon}||\gamma(t(u))||$.*

This theorem justifies that, despite the kernel's Frobenius norm changing, the training dynamics can still be accurately approximated using the initial kernel. It allows us to use the NTK calculated at the initial state of an LLM as a reliable proxy for data utility throughout the fine-tuning process,

which is the core principle behind our data selection method detailed in §4.

## 3.2. Jacobian-free Approximation.

Computing the exact NTK as defined in Equation (2) for LLMs is computationally prohibitive, as it requires constructing and storing the full Jacobian matrix. This is infeasible given the immense dimensionality of both model parameters and outputs. To overcome this challenge, we introduce a scalable, Jacobian-free NTK approximation.

**Definition 3.3** (Jacobian-free NTK Approximation). Given a model $f(\cdot; \theta_t) : \mathbb{R}^d \to \mathbb{R}^D$, the Jacobian-free NTK approximation between two inputs $\mathbf{x}$ and $\mathbf{x}'$ is defined as:

$$\widetilde{\Theta}(\mathbf{x}, \mathbf{x}'; \theta_t) = \left\langle \nabla_{\theta_t} \sum_{k=1}^{D} f_k(\mathbf{x}; \theta_t), \nabla_{\theta_t} \sum_{k=1}^{D} f_k(\mathbf{x}', \theta_t) \right\rangle,$$
(5)

This approximation can be expanded to reveal its relationship with the exact NTK: $\widetilde{\Theta}(\mathbf{x}, \mathbf{x}'; \theta_t) = \sum_{k=1}^{D} \langle \nabla_{\theta_t} f_k(\mathbf{x}; \theta_t), \nabla_{\theta_t} f_k(\mathbf{x}'; \theta_t) \rangle + \sum_{k=1}^{D} \sum_{m=1, m \neq k}^{D} \langle \nabla_{\theta_t} f_k(\mathbf{x}; \theta_t), \nabla_{\theta_t} f_m(\mathbf{x}', \theta_t) \rangle$, whose concrete derivation is provided in Appendix D. The first term precisely recovers the exact NTK, while the second term accounts for cross-output interactions. In practice, we empirically observe that the gradient directions for different output dimensions tend to be nearly orthogonal, making the cross terms very small. As shown in Figure 2b, we visualize the probability distribution of gradient similarity between cross outputs, demonstrating that the cross outputs exhibit rather low similarity ($< 0.02$). Results on more tasks and the derived kernel similarity are demonstrated in Appendix H.2. We also provide a theoretical analysis in Appendix E that explains this empirical behavior and goes beyond prior NTK analyses limited to idealized conditions (Mohamadi & Sutherland, 2022). This confirms that the relative magnitudes of NTK values are well-preserved, which provides a reliable proxy for the exact kernel and justifies our use of $\widetilde{\Theta}$ for efficient data selection.

# 4. Method

## 4.1. Total Framework

Based on the above insights, we propose NTK-Selector, a novel two-stage data selection framework. Given a domain dataset $\mathcal{D}$, candidate dataset $\mathcal{C}$, pre-selection size $M$, number of nearest neighbors $K$, LLM $f(\cdot; \theta)$, and random projection dimension matrix $\Pi$, our goal is to efficiently select a high-value subset $\mathcal{S} \subset \mathcal{C}$. The process is outlined in Algorithm 1 and contains two main phases: (i) **Coarse-grained pre-selection** (§4.2): an efficient embedding-based filtering step that constructs a reduced candidate set $\mathcal{S}_{\text{pre}} \subset \mathcal{C}$; (ii) **Fine-grained NTK selection** (§4.3): a computationally

---

**Algorithm 1** NTK-Selector

**Input:** $\mathcal{D}, \mathcal{C}, M, K \ f(\cdot; \theta), \Pi$.
**Output:** $\mathcal{S}$.
1: $\mathcal{S}_{\text{pre}} \leftarrow \text{PRE-SELECT}(\mathcal{D}, \mathcal{C}, M, K, f(\cdot; \theta))$;
2: **for** $\mathbf{x} \in \mathcal{D} \cup \mathcal{S}_{\text{pre}}$ **do**
3:     Compute LoRA gradient: $\widehat{\nabla}_\theta f(\mathbf{x}; \theta) = \nabla_{\theta_{\text{LoRA}}} \left( \sum_{k=1}^{D} f_k(\mathbf{x}; \theta) \right)$;
4:     Project gradient: $\widetilde{\nabla}_\theta f(\mathbf{x}; \theta) = \Pi^\top \widehat{\nabla}_\theta f(\mathbf{x}; \theta)$;
5: **end for**
6: **for** $i = 1, \dots |\mathcal{D}|$ **do**
7:     **for** $j = 1, \dots M$ **do**
8:         $\widetilde{\Theta}(\mathbf{x}_i, \mathbf{x}_j; \theta) \leftarrow \left\langle \widetilde{\nabla}_\theta f(\mathbf{x}_i; \theta), \widetilde{\nabla}_\theta f(\mathbf{x}_j; \theta) \right\rangle$ where $\mathbf{x}_i \in \mathcal{D}, \mathbf{x}_j \in \mathcal{S}_{\text{pre}}$;
9:     **end for**
10: **end for**
11: Compute NTK utility score: $s_j \leftarrow \frac{1}{|\mathcal{D}|} \sum_{\mathbf{x}_i \in \mathcal{D}} \widetilde{\Theta}(\mathbf{x}_i, \mathbf{x}_j; \theta)$ for $j = 1, \dots, M$;
12: Select Top-$N$ samples: $\mathcal{S} \leftarrow \{\text{Top-}N \text{ samples from } \mathcal{S}_{\text{pre}} \text{ ranked by their score } s_j\}$.

---

refined stage to select $\mathcal{S}$ involving gradient computation, random projection, and Jacobian-free NTK approximation to estimate sample utility over the pre-selected set $\mathcal{S}_{\text{pre}}$.

## 4.2. Coarse-grained Pre-selection

The computational burden of evaluating the NTK over a very large candidate set $\mathcal{C}$ (where $|\mathcal{C}| \gg 10^6$) necessitates an efficient pre-selection mechanism. We thus propose a two-stage procedure, beginning with a coarse-grained filtering step based on embedding similarity, to reduce the candidate set to a tractable size prior to fine-grained NTK assessment. Let $\phi : \mathcal{X} \to \mathbb{R}^m$ denote a feature mapping obtained by averaging the hidden states of the final transformer layer. To better capture domain-specific contextual nuances, we perform a warm-up LoRA training phase on the available domain data $\mathcal{D}$ before computing the embeddings. For a domain example $\mathbf{x}_i \in \mathcal{D}$ and candidate $\mathbf{x}_j \in \mathcal{C}$, let $\mathbf{d}_i = \phi(\mathbf{x}_i)$ and $\mathbf{c}_j = \phi(\mathbf{x}_j)$ be their respective embeddings. We define the Euclidean distance between embeddings as $d(\mathbf{d}_i, \mathbf{c}_j) = \|\mathbf{d}_i - \mathbf{c}_j\|_2$. For each $\mathbf{d}_i$, we identify the set $N_K(\mathbf{d}_i)$ of indices corresponding to the $K$ nearest neighbors in $\{\mathbf{c}_j\}_{j=1}^{|\mathcal{C}|}$ under this metric. The relevance of a candidate $\mathbf{x}_j$ to the entire domain set $\mathcal{D}$ is quantified by the number of domain points for which it ranks among the top-$K$ neighbors: $r(\mathbf{x}_j) = \sum_{i=1}^{|\mathcal{D}|} \mathbf{1} \left( j \in N_K(\mathbf{d}_i) \right)$, where $\mathbf{1}(\cdot)$ is the indicator function. The pre-selected set $\mathcal{S}_{\text{pre}} \subset \mathcal{C}$ of size $M$ is then constructed by taking the candidates with the highest relevance scores: $\mathcal{S}_{\text{pre}} = \text{Top}_M(r(\mathbf{x}))$. This $\mathbf{x} \in \mathcal{C}$

embedding-based pre-selection serves as a scalable proxy for semantic relevance, significantly reducing the number of samples while preserving high-utility candidates.

### 4.3. Fine-grained NTK Selection

Given the pre-selected set $\mathcal{S}_{\text{pre}}$, we now proceed with the fine-grained selection using our NTK-based utility score. We define the NTK utility score $s_j$ for a candidate point $\mathbf{x}_j \in \mathcal{S}_{\text{pre}}$ as its average NTK similarity to all points in the domain dataset $\mathcal{D}$: $s_j = \frac{1}{|\mathcal{D}|} \sum_{\mathbf{x}_i \in \mathcal{D}} \widetilde{\Theta}(\mathbf{x}_i, \mathbf{x}_j; \theta)$, where $\widetilde{\Theta}$ is our Jacobian-free NTK approximation as described in §3.2. As justified in §2.2 and §3.1, selecting data points that maximize this utility score serves as a practical surrogate for the intractable bi-level objective function in Equation (1). This greedy maximization strategy, which selects $N$ candidates with the highest scores, avoids the need for expensive, iterative sample enumeration, training, and evaluation for every candidate subset, providing a scalable and effective solution for selecting relevant auxiliary data.

**Parameter Efficient NTK via LoRA.** To alleviate the memory pressure of gradient computations, we leverage a low-rank adaptation (LoRA) (Hu et al., 2022) to restrict gradient estimation to a low-rank parameter subspace. By focusing on the gradients of the LoRA modules, which are denoted as $\widehat{\nabla}_\theta f(\cdot; \theta) := \nabla_{\theta_{\text{LoRA}}} f(\cdot; \theta) \in \mathbb{R}^P$ with $P$ LoRA parameters, we can work within a much lower-dimensional parameter space compared to the full model. For instance, in LLAMA3-8B-INSTRUCT, the gradient dimensionality of the LoRA modules is only about 0.5% of the full model's gradient space. This significant reduction enables efficient and scalable NTK approximation without compromising empirical performance.

**Random Projection for Scalable NTK Estimation.** To scale our NTK approximation to large candidate sets, we apply random projection to the LoRA gradients, further reducing memory and computational requirements (Park et al., 2023). This approach is motivated by the Johnson-Lindenstrauss Lemma (Johnson et al., 1984), which guarantees that inner products are approximately preserved under low-dimensional random projections. Formally, for each input $\mathbf{x}$ we project its LoRA gradient $\widehat{\nabla}_\theta f(\cdot; \theta) \in \mathbb{R}^P$ into a $p$-dimensional space ($p \ll P$) via $\widetilde{\nabla}_\theta f(\cdot; \theta) = \Pi^\top \widehat{\nabla}_\theta f(\cdot; \theta)$, where $\Pi \in \mathbb{R}^{P \times p}$ is a random projection matrix with entries drawn independently from a Rademacher distribution ($\Pi_{i,j} = \pm 1$ with equal probability ). The same $\Pi$ is applied consistently across all samples using a fixed random seed, ensuring that the approximated inner products used in Equation (5) remain consistent and valid for calculation of the utility score. We evaluate the impact of random projection in Appendix H.3.

## 5. Experiments

### 5.1. Setup

**Datasets.** We evaluate on four low-resource domains: medical (**MedMCQA** (Pal et al., 2022), **MMLU-Med** (Hendrycks et al., 2021)), financial (**FPB** (Malo et al., 2014), **TFNS** (Magic, 2022), **Headline** (Sinha & Khandait, 2021)), legal (**ContractNLI** (Koreeda & Manning, 2021), **PrivacyQA** (Ravichander et al., 2019)), and psychological (**RSDD** (Yates et al., 2017)) domains. Additional dataset details are provided in Appendix G.1. For candidate data, we use the **Cot Collection** (Kim et al., 2023) (1.8M instruction-response pairs) for auxiliary data selection. Domain-specific training sets are used as instruction datasets when applicable, except for the medical domain, for which we use **UltraMedical** (Zhang et al., 2024) for instruction tuning. For training instances lacking chain-of-thought responses, we generate such responses using GPT-4o-mini (Hurst et al., 2024) (see reasons in Appendix H.9). Further examples of query templates and generated responses are included in Appendix I.

**Baselines.** We compare NTK-Selector against the following baselines to evaluate its effectiveness. The simplest is **Base**, which assesses the out-of-the-box performance of the pre-trained backbone model without any fine-tuning. Another baseline, **Domain-Only**, involves fine-tuning the model exclusively on a limited amount of domain-specific data without augmentation. For selecting auxiliary data, we explore several strategies. The most straightforward is **Random** selection, where general-purpose data is randomly sampled and mixed with the domain-specific dataset. This baseline helps gauge the benefit of simply adding more data. Besides, we implement selection based on **Embedding** and **Gradient** similarity, which is calculated by the averaged last hidden state and loss gradients, respectively. We also compare with **DSIR** (Xie et al., 2023), which uses $n$-gram features to weight candidate data and samples based on these weights. **LESS** (Xia et al., 2024) is another baseline that selects data by approximating the first-order influence of each candidate sample on the validation set. Lastly, we use **TSDS** (Liu et al., 2024), which captures the discrepancy between the candidate and target data distributions by aligning them using optimal transport. Please refer to Appendix G.2 for more implementation details.

**Models & Settings.** We employ **Llama3-8B-Instruct** (Dubey et al., 2024) and **QWEN3-8B** (Yang et al., 2025) as base models. All fine-tuning is performed using LoRA. We randomly sample 1K instances from each of the domain datasets for augmentation, and select 9K instances from the candidate pool for mixed training. Models are mixed training for 3 epochs, and the gradients are randomly projected to 8192-dimensional features. The pre-selection stage is implemented using Faiss (Douze et al., 2024)

*Table 1.* Performance of LLAMA3-8B-INSTRUCT and QWEN3-8B across 8 tasks on medical (MedMCQA and MMLU-Med), financial (FPB, TFNS, and Headline), legal (ContractNLI and PrivacyQA), and psychological (RSDD) domains. The best individual task results for each model are highlighted in **bold**, and relative performance changes compared to the Base model are denoted by ↑ (increase), ↓ (decrease), and → (no change).

| | Medical | | Financial | | | Legal | | Psycho. | |
| | MedMCQA | MMLU-Med | FPB | TFNS | Headline | ContractNLI | PrivacyQA | RSDD | Avg. |
|---|---|---|---|---|---|---|---|---|---|
| | | | | LLAMA3-8B-INSTRUCT | | | | | |
| Base | 56.7 | 72.3 | 81.7 | 57.2 | 78.5 | 59.9 | 57.9 | 78.8 | 67.9 |
| Domain-Only | 56.5 $_{0.2\downarrow}$ | 71.8 $_{0.5\downarrow}$ | 80.1 $_{1.6\downarrow}$ | 69.3 $_{12.1\uparrow}$ | 82.4 $_{3.9\uparrow}$ | 41.1 $_{18.8\downarrow}$ | 63.6 $_{5.7\uparrow}$ | 85.1 $_{6.3\uparrow}$ | 68.7 |
| Random | 57.1 $_{0.4\uparrow}$ | 73.0 $_{0.7\uparrow}$ | 80.6 $_{1.1\downarrow}$ | 70.7 $_{13.5\uparrow}$ | 85.2 $_{6.7\uparrow}$ | 66.5 $_{6.6\uparrow}$ | 66.2 $_{8.3\uparrow}$ | 95.4 $_{16.6\uparrow}$ | 74.3 |
| Embedding | 57.4 $_{0.7\uparrow}$ | 72.7 $_{0.3\uparrow}$ | 86.0 $_{4.3\uparrow}$ | 71.4 $_{14.2\uparrow}$ | 83.7 $_{5.2\uparrow}$ | 67.7 $_{7.8\uparrow}$ | 66.2 $_{8.3\uparrow}$ | 94.4 $_{15.6\uparrow}$ | 74.9 |
| Gradient | 57.4 $_{0.7\uparrow}$ | 74.0 $_{1.6\uparrow}$ | 86.0 $_{4.3\uparrow}$ | 68.9 $_{11.7\uparrow}$ | 85.4 $_{6.9\uparrow}$ | 61.6 $_{1.7\uparrow}$ | 62.9 $_{5.0\uparrow}$ | 94.6 $_{15.8\uparrow}$ | 73.9 |
| DSIR | 57.5 $_{0.8\uparrow}$ | 72.7 $_{0.4\uparrow}$ | 84.6 $_{2.9\uparrow}$ | 71.2 $_{14.0\uparrow}$ | **86.2** $_{7.7\uparrow}$ | 64.5 $_{4.6\uparrow}$ | 66.5 $_{8.6\uparrow}$ | 94.7 $_{15.9\uparrow}$ | 74.7 |
| LESS | 57.5 $_{0.8\uparrow}$ | 73.1 $_{0.8\uparrow}$ | 82.4 $_{0.7\uparrow}$ | 70.7 $_{13.5\uparrow}$ | 85.9 $_{7.4\uparrow}$ | 67.8 $_{7.9\uparrow}$ | 66.6 $_{8.7\uparrow}$ | 94.7 $_{15.9\uparrow}$ | 74.8 |
| TSDS | 57.0 $_{0.3\uparrow}$ | 71.4 $_{0.9\downarrow}$ | 83.3 $_{1.6\uparrow}$ | 71.2 $_{14.0\uparrow}$ | 85.0 $_{6.5\uparrow}$ | 68.9 $_{9.0\uparrow}$ | 66.3 $_{8.4\uparrow}$ | 94.9 $_{16.1\uparrow}$ | 74.8 |
| **NTK-Selector** | **59.1** $_{2.4\uparrow}$ | **73.8** $_{1.5\uparrow}$ | **85.5** $_{3.8\uparrow}$ | **73.3** $_{16.1\uparrow}$ | **86.2** $_{7.7\uparrow}$ | **70.6** $_{10.7\uparrow}$ | **67.8** $_{9.9\uparrow}$ | **96.5** $_{17.7\uparrow}$ | **76.6** |
| | | | | QWEN3-8B | | | | | |
| Base | 59.3 | 83.4 | 80.1 | 70.2 | 74.0 | 73.8 | 52.3 | 94.6 | 73.5 |
| Domain-Only | 61.1 $_{1.8\uparrow}$ | 81.9 $_{1.5\downarrow}$ | 72.2 $_{7.9\downarrow}$ | 72.0 $_{1.8\uparrow}$ | 75.3 $_{1.3\uparrow}$ | 76.7 $_{2.9\uparrow}$ | 66.5 $_{14.2\uparrow}$ | 89.4 $_{5.2\downarrow}$ | 74.4 |
| Random | 60.8 $_{1.5\uparrow}$ | 82.6 $_{0.8\downarrow}$ | 65.2 $_{14.9\downarrow}$ | 71.9 $_{1.7\uparrow}$ | 80.9 $_{6.9\uparrow}$ | 77.3 $_{3.5\uparrow}$ | 53.9 $_{1.6\uparrow}$ | 90.8 $_{3.8\downarrow}$ | 72.9 |
| Embedding | 60.9 $_{1.6\uparrow}$ | 81.5 $_{1.9\downarrow}$ | 85.2 $_{5.1\uparrow}$ | 66.7 $_{3.5\downarrow}$ | 81.8 $_{7.8\uparrow}$ | 79.3 $_{5.5\uparrow}$ | 53.3 $_{1.0\uparrow}$ | 93.9 $_{0.7\downarrow}$ | 75.3 |
| Gradient | 60.7 $_{1.4\uparrow}$ | 82.2 $_{1.2\downarrow}$ | 82.5 $_{2.4\uparrow}$ | 67.9 $_{2.3\downarrow}$ | 82.3 $_{8.3\uparrow}$ | 75.2 $_{1.4\uparrow}$ | 66.1 $_{13.8\uparrow}$ | 91.6 $_{3.0\downarrow}$ | 76.1 |
| DSIR | 61.1 $_{1.8\uparrow}$ | 82.7 $_{0.7\downarrow}$ | **85.9** $_{5.8\uparrow}$ | 69.2 $_{1.0\downarrow}$ | 82.3 $_{8.3\uparrow}$ | 78.8 $_{5.0\uparrow}$ | 67.1 $_{14.8\uparrow}$ | 94.6 $_{0.0\rightarrow}$ | 77.7 |
| LESS | 60.3 $_{1.0\uparrow}$ | 82.6 $_{0.8\downarrow}$ | 60.3 $_{19.8\downarrow}$ | 69.8 $_{0.4\downarrow}$ | 81.8 $_{7.8\uparrow}$ | 78.6 $_{4.8\uparrow}$ | 62.3 $_{10.0\uparrow}$ | 93.8 $_{0.8\downarrow}$ | 73.7 |
| TSDS | 60.6 $_{1.3\uparrow}$ | 82.3 $_{1.1\downarrow}$ | 84.7 $_{4.6\uparrow}$ | 68.1 $_{2.1\downarrow}$ | 81.9 $_{7.9\uparrow}$ | 79.8 $_{6.0\uparrow}$ | 50.1 $_{2.2\downarrow}$ | 92.5 $_{2.1\downarrow}$ | 75.0 |
| **NTK-Selector** | **61.4** $_{2.1\uparrow}$ | **83.8** $_{0.4\uparrow}$ | 85.3 $_{5.2\uparrow}$ | 72.2 $_{2.0\uparrow}$ | **83.4** $_{9.4\uparrow}$ | **79.9** $_{6.1\uparrow}$ | **67.5** $_{15.2\uparrow}$ | **95.1** $_{0.5\uparrow}$ | **78.6** |

library with $M = 4N$, and $K = M/4$ for coarse-grained pre-selection. To eliminate the effect of the sequence length of each data point, we normalize the gradient in Equation (5) with the sequence length. For numerical stability, gradient sums are scaled by a factor of $10^{-5}$. For the main comparisons, NTK-Selector and all baselines train from scratch on their respective selected datasets, which means they share the same initialization, training procedure, and hyperparameter settings, and differ only in the training data. Additional training hyperparameters are detailed in Appendix G.3 and G.4.

### 5.2. Main Results

Table 1 presents the main results of our approach and various baselines across different models and target domains. All results are averaged over 3 runs. Below, we summarize our key findings.

**Limited domain data can degrade performance.** Fine-tuning with only 1K domain-specific data points yielded surprising results. Instead of consistent improvement, we observed that this limited data volume often led to a negligible performance gain or, in some cases, a degradation in task capability (e.g., LLAMA3-8B-INSTRUCT drops 18.8 points on ContractNLI; QWEN3-8B drops 7.9 points on

FPB). We attribute this phenomenon to overfitting (Béthune et al., 2025), where the fine-tuning process overwrites the extensive general knowledge acquired during pre-training. Furthermore, the small dataset size fails to provide a stable optimization landscape, leading to training instability, poor convergence, and a tendency for the model to overfit to noise within the data. Together, these effects underscore the need for auxiliary supervision beyond domain-only training.

**Auxiliary data can usually, but not necessarily, boost the model performance.** We selected 9K auxiliary samples using various methods and combined them with the 1K target data points for mixed training. For LLAMA3-8B-INSTRUCT, even random selection improves performance across most domains. In contrast, for QWEN3-8B, random selection often degrades performance, for instance, reducing gains on PrivacyQA from 14.2 to just 1.6 points. This negative effect is even further exacerbated by some advanced selection methods (e.g., LESS induces a 49.8-point drop on FPB). We hypothesize that this difference stems from QWEN3-8B's superior initial performance, which makes it more sensitive to the quality of auxiliary data, leading low-quality samples to more easily disrupt the training process of a powerful model.

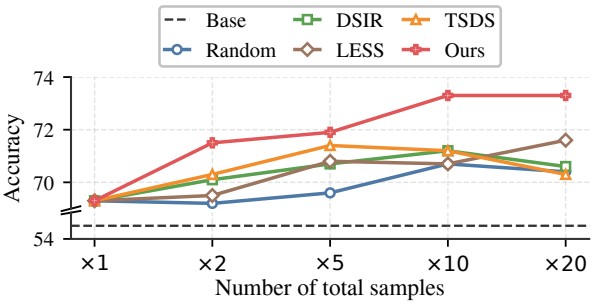

*Figure 3.* Accuracy on TFNS task with 1K domain samples and various numbers of auxiliary samples, where the total number is enriched by ×2, ×5, ×10, and ×20.

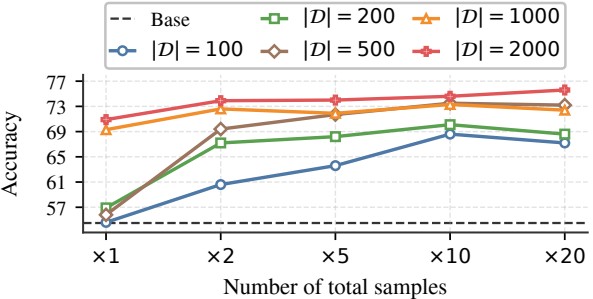

*Figure 4.* Accuracy on TFNS task with 100, 200, 500, 1K, 2K domain samples enriched by ×2, ×5, ×10, and ×20.

**NTK-Selector is the only consistently effective approach.** While conventional methods like DSIR ($n$-gram features), LESS (influence functions), and TSDS (representation alignment) can be effective for coreset selection, they often fail to provide stable improvements when selecting auxiliary data from a general, out-of-domain corpus. In contrast, NTK-Selector consistently enhances model performance across all domains and architectures. Our approach selects auxiliary data whose optimization trajectories align with those of the target domain, thereby stabilizing training and promoting more robust convergence. As shown in Table 1, NTK-Selector achieves the highest average performance on both LLAMA3-8B-INSTRUCT (76.6, from a base of 67.9) and QWEN3-8B (78.6, from 73.5). Compared to the domain-only baseline, which improves by only 0.8 and 0.9 points respectively, our method delivers gains of 8.7 and 5.3 points, corresponding to a **10.9x and 5.7x improvement**. Notably, NTK-Selector is the only method that improves performance on every task, including MMLU-Med and RSDD for QWEN3-8B, where other techniques often degrade results. This demonstrates that NTK-Selector reliably identifies high-value auxiliary examples that complement the target domain, leading to superior and generalizable gains.

## 6. Analysis

This section presents an analysis of the NTK-Selector framework through multiple ablations. We first examine how the quantity of auxiliary data affects performance, and how the size of the target dataset influences the utility of added auxiliary examples. We then ablate key hyperparameters to provide practical recommendations, such as pre-selection size and projection dimension. Finally, we compare the performance of our method against empirical NTK regression to further validate the NTK-like behavior of LLMs. Unless specified, all experiments use LLAMA3-8B-INSTRUCT with 1K domain examples and 9K auxiliary samples. More analyses are listed in Appendix H.4 to H.11.

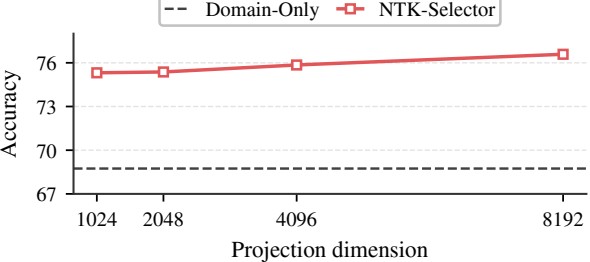

*Figure 5.* Average performance of NTK-Selector using different projected dimensions.

**Scaling behavior with auxiliary data** We investigate how the volume of auxiliary data affects performance by holding the domain set fixed at 1K samples (×1) and scaling the total data size to ×2, ×5, ×10, and ×20 via added auxiliary examples. As shown in Figure 3, NTK-Selector consistently outperforms all baselines at every scale. Notably, the performance gains exhibit clear diminishing returns: while increasing data from ×1 to ×5 brings substantial gains (69.3→71.9), further expanding to ×10 and ×20 yields minimal improvement (73.3). We hypothesize that the supply of highly relevant cross-domain samples is inherently limited. Beyond a certain point (e.g., ×10), incorporating less relevant data contributes little additional signal and may even dilute fine-tuning efficacy, suggesting the existence of a practical saturation threshold for auxiliary data utility. This observation emphasizes the importance of sample quality over sheer quantity in data selection.

**Less target data, more pronounced improvement with auxiliary data.** To further analyze the effectiveness of auxiliary data, we explored its impact on different volumes of target data, ranging from a very small set $|\mathcal{D}| = 100$ to a larger one $|\mathcal{D}| = 2000$. As shown in Figure 4, the performance of the baseline (×1) predictably improves as the amount of target data increases, and the results demonstrate that our method consistently provides significant gains across all domain data sizes. Crucially, we found that the

*Table 2.* Accuracy with size $M$ set as $2N$, $4N$ (default), and $16N$ in the coarse-grained pre-selection stage.

|             | MMLU-Med | FPB | ContractNLI | RSDD |
|-------------|----------|-----|-------------|------|
| Base        | 72.3     | 81.7 | 59.9 | 78.8 |
| Domain-Only | 71.8 $_{0.5\downarrow}$ | 80.1 $_{1.5\downarrow}$ | 41.1 $_{18.8\downarrow}$ | 85.1 $_{6.3\uparrow}$ |
| $M = 2N$    | 73.3 $_{1.0\uparrow}$ | 84.5 $_{2.8\uparrow}$ | 68.9 $_{9.0\uparrow}$ | 95.5 $_{16.7\uparrow}$ |
| $M = 4N$    | 73.7 $_{1.4\uparrow}$ | 85.5 $_{3.8\uparrow}$ | **70.6** $_{10.7\uparrow}$ | **96.5** $_{17.7\uparrow}$ |
| $M = 16N$   | **74.4** $_{2.1\uparrow}$ | **86.9** $_{5.2\uparrow}$ | **70.6** $_{10.7\uparrow}$ | 95.9 $_{17.1\uparrow}$ |

*Table 3.* The fine-tuning (FT.) and eNTK accuracy on different domains. ✓ denotes that the kernel analog achieves >90% of FT. accuracy.

|       | Fin. | Legal | Psycho. |
|-------|------|-------|---------|
| FT.   | 77.27 | 52.35 | 85.10 |
| eNTK  | 71.57 | 54.45 | 88.40 |
| $\Delta$ | 5.70 ✓ | 2.10 ✓ | 3.30 ✓ |

*Table 4.* Asymptotic complexity and wall-clock runtime (measured as single NVIDIA A100 GPU minutes) with each step in NTK-Selector.

|            | Coarse-grained Pre-selection | | | Fine-grained NTK Selection | |
|------------|------------------|----------------------|-----------------|----------------------|----------------|
|            | Warm-up training | Embedding Computation | KNN Selection | Gradient Computation | NTK selection |
| Complexity | $\mathcal{O}(\|\mathcal{D}\|)$ | $\mathcal{O}(\|\mathcal{C}\|)$ | $\mathcal{O}(\|\mathcal{C}\| \cdot M)$ | $\mathcal{O}(M)$ | $\mathcal{O}(M \cdot N)$ |
| Runtime    | 8 Min | 0.9 Min/K samples | 10 Min | 6 Min/K samples | 1 Min |

relative gain from auxiliary data is most pronounced in the most resource-constrained settings. For example, with only $|\mathcal{D}| = 100$ samples, performance improves from 54.6 to 68.6 at the $\times 10$ scale, which results in a gain of 14.0 points (25.6%). In contrast, with $|\mathcal{D}| = 2000$, the improvement is only from 70.9 to 74.6, a gain of 3.7 points (5.2%). It empirically demonstrates that NTK-Selector is especially powerful in very low-resource scenarios, where it compensates effectively for the lack of in-domain data.

**Small projection dimension is sufficient.** This part investigates the influence of the gradient projection dimension $p$, which varies across 1024, 2048, 4096, and 8192 (default). As shown in Figure 5, even the smallest dimension ($p$=1024) yields significant improvement over the domain-only baseline. Performance increases steadily with larger dimensions, suggesting that higher-dimensional projections better preserve gradient similarity information. Therefore, we recommend using a larger projection dimension if computational resources allow.

**Scaling pre-selection.** To enhance scalability over large candidate sets, NTK-Selector employs a coarse-grained pre-selection stage that reduces the candidate pool from millions to a manageable set of $M$ samples. We evaluate the effect of $M$ in Table 2 using multiples of the final subset size $N$: $2N$, $4N$ (default), and $16N$. The results confirm that a larger pre-selection size consistently improves accuracy across tasks. For example, increasing $M$ from $2N$ to $16N$ brings substantial gains in MMLU-Med ($73.3 \rightarrow 74.4$) and FPB ($84.5 \rightarrow 86.9$), with more moderate improvements in ContractNLI and RSDD. However, larger values of $M$ also entail higher computational and memory costs during the subsequent NTK scoring stage. We therefore recommend choosing $M$ within the range of $4N$ to $16N$, which provides a practical trade-off between selection quality and efficiency.

**Empirical NTK regression.** To further validate the NTK-like behavior of pre-trained LLMs, we compare the fine-tuning accuracy against predictions from empirical Neural Tangent Kernel (eNTK) regression following (Wei et al., 2022) and (Malladi et al., 2023). For each task, we compute the eNTK kernel matrix and solve the associated kernel regression problem (see Appendix F for implementation details). As shown in Table 3, eNTK regression achieves 90% of fine-tuning accuracy in all cases across financial, legal, and psychological domains. This close agreement provides further evidence that the gradient-based dynamics of LLMs during task-specific fine-tuning are well-approximated by a static kernel, supporting our observation in §3.1. These results reinforce the practical relevance of NTK-based analysis and selection mechanisms for LLMs.

**Computational complexity and runtime analysis.** Table 4 outlines the asymptotic complexity and wall-clock runtime for each stage of NTK-Selector in Algorithm 1. The wall-clock time is measured in single NVIDIA A100 (80GB) GPU minutes. The most computationally expensive steps are embedding and gradient computation, which scale linearly with the candidate size $|\mathcal{C}|$ and the pre-selection size $M$. Under our default configuration, which selects $N = 9000$ samples from a candidate pool of 1.8 million samples with $M = 4N$, the total runtime is approximately 31 hours. This cost scales favorably and can be substantially reduced by using smaller candidate sets or pre-selection sizes, making the method practical for large-scale data selection. It is noted that the warm-up stage is only applied to the model used for data selection, not to the model used for final performance evaluation.

# 7. Conclusion

In this paper, we propose NTK-Selector, a principled framework for selecting high-value auxiliary data from general corpora to address data scarcity in low-resource domains. By leveraging the observation that LLMs exhibit stable NTK-like behavior during fine-tuning, NTK-Selector enables efficient and theoretically motivated data selection. Extensive experiments show that NTK-Selector consistently outperforms existing data selection methods across multiple models and domains, with particularly strong gains in very low-resource settings. Our work establishes a new SOTA in cross-domain data selection and offers deeper insights into the optimization dynamics of LLMs during fine-tuning.

# Acknowledgement

This work was supported by the National Natural Science Foundation of ChinaGrant No. 62576209) and the Shanghai Municipal Special Program for Basic Research on General AI Foundation Models (Grant No. 2025SHZDZX025G05).

# Impact Statement

This paper presents a methodological contribution aimed at improving low-resource domain adaptation for large language models. By enabling the effective selection of auxiliary data from general-domain corpora, our approach may help reduce reliance on scarce, expensive, or sensitive domain-specific data, thereby facilitating broader access to language model capabilities in specialized application areas. The proposed method does not involve new data collection, model deployment, or the use of additional sensitive information, and its ethical considerations are therefore aligned with those commonly associated with research on large language models. No specific negative societal impacts are anticipated beyond these well-established considerations.

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

## A. The Use of Large Language Models

We leveraged large language models as writing assistants for tasks such as rephrasing sentences, improving grammatical flow, and refining technical descriptions for clarity.

## B. Related Works

**Task-agnostic Data Selection.**  Conventional data selection approaches are broadly classified into two main categories: task-agnostic and task-specific methods. Task-agnostic methods aim to curate a high-quality and representative subset of data by filtering out low-quality and redundant samples, without a specific target task in mind. These methods can be further subdivided into three classes: (i) Quality-based approaches often utilize the training loss or other proxy metrics to score and select data. For example, PDS (Gu et al., 2025) formulates data selection as an optimal control problem and solves it using Pontryagin's Maximum Principle theory (Pontryagin, 2018) to optimize the AUC under the training loss curve, while IFD (Li et al., 2024b) measures data quality by calculating the loss difference when removing the instruction input. (ii) Diversity-based algorithms focus on reducing dataset redundancy. They measure data diversity using metrics such as feature distance (Tirumala et al., 2023), eigenvalues (Fan et al., 2025), and gradient similarity (Zhang et al., 2025). (iii) Model-based methods employ a proxy model, which can be a small model or a powerful large language model (e.g., ChatGPT (OpenAI, 2023)), to rate each candidate sample. Notable examples include AlpaGasus (Chen et al., 2024), SuperFiltering (Li et al., 2024a), and ADO (Jiang et al., 2025). A key limitation of these task-agnostic approaches is their inability to provide targeted data augmentation for a specific domain or downstream task, as they do not rely on a reference dataset to guide the selection process.

**Task-specific Data Selection.**  In contrast to task-agnostic methods, task-specific approaches focus on curating a subset of data from a large in-domain candidate pool specifically for a target task. These methods often leverage sophisticated techniques to identify the most relevant data points. For instance, LESS (Xia et al., 2024) utilizes a modified influence function tailored for the Adam optimizer, achieving high accuracy with as little as 5% of the full dataset. Similarly, G-DIG (Pan et al., 2024), which is focused on machine translation, also adopts an influence function-based approach guided by a manually curated seed subset. Other methods formulate the problem as an optimization task. TSDS (Liu et al., 2024) frames data selection for task-specific finetuning as an optimization problem that minimizes the discrepancy between the selected data and the target distribution using an optimal transport-based distribution alignment loss. Alternatively, DSIR (Xie et al., 2023) bypasses the need for model features or gradients altogether. It instead constructs an importance weight estimator using $n$-gram features to evaluate the relevance of each candidate sample. While these approaches have shown effectiveness, they all fundamentally rely on a substantial quantity of in-domain candidate pool and a reference validation set. They do not address scenarios with extremely limited resources or investigate how leveraging out-of-domain data could benefit model training and convergence.

**Neural Tangent Kernel.**  Neural Tangent Kernel (Jacot et al., 2018b) is a foundational concept demonstrating that training an neural network under certain parameterization is equivalent to performing kernel regression as the width of the network approaches infinity. This insight has become a cornerstone for understanding the generalization properties of deep networks. Following this seminal work, subsequent research (Allen-Zhu et al., 2019; Cao & Gu, 2019; Wei et al., 2022) has widely applied the NTK lens to analyze the optimization and generalization behavior of deep learning models. However, this theoretical perspective has largely been confined to small, randomly initialized networks to satisfy the core assumptions of the NTK framework. There has been little investigation into its application within the pre-training and fine-tuning paradigm, especially for large language models (LLMs). While some recent works, such as those by (Malladi et al., 2023) and (Afzal et al., 2024), have explored the kernel behavior of pre-trained language models (e.g., RoBERTa (Liu et al., 2019)) on the GLUE benchmark (Wang et al., 2019), these studies are still limited to small-scale models and classification tasks. This is far from the real-world applications of modern LLMs. In this paper, we present empirical evidence for stable NTK-like behavior in LLMs, and we leverage this observation to design an NTK-based auxiliary data selection criterion that yields consistent enhancement for low-resource domains. To our knowledge, this is the first systematic study of kernel behavior in large language models under standard instruction tuning.

# C. Proof of Theorem 3.2

**Theorem C.1.** *Given a model $f(\cdot; \theta_t)$ with NTK $\Theta(\cdot, \cdot; \theta_t)$ whose training dynamics are governed by a gradient flow $\dot{f}(\cdot; \theta_t) = -\eta\Theta(\cdot, \cdot; \theta_t)\gamma(t)$, where $\gamma(t) = \nabla_f \mathcal{L}(f(\cdot; \theta_t))$ and $\eta$ is the learning rate, if it exhibits NTK-like behavior on $[0, T]$, its training dynamics can be re-parameterized onto a new time axis $u$:*

$$\dot{f}(\cdot; \theta_{t(u)}) = -\eta\Theta(\cdot, \cdot; \theta_0)\gamma(t(u)) + \Delta(u), \tag{6}$$

*where $u(t) := \int_0^t a^*(\tau)d\tau$, $a^*(t) = \frac{\langle\Theta(\cdot,\cdot;\theta_t),\Theta(\cdot,\cdot;\theta_0)\rangle}{||\Theta(\cdot,\cdot;\theta_0)||^2}$ and the perturbation term $\Delta(u)$ is bounded with $||\Delta(u)|| \leq \eta||\Theta(\cdot,\cdot;\theta_0)||\frac{\sqrt{2\epsilon}}{1-\epsilon}||\gamma(t(u))||$.*

*Proof.* For brevity, we let $\dot{f}(t) := \dot{f}(\cdot; \theta_t)$, $\dot{f}(u) := \dot{f}(\cdot; \theta_{t(u)})$, $\Theta(t) := \Theta(\cdot, \cdot; \theta_t)$ and $\Theta_0 := \Theta(0)$. We decompose $\Theta(t)$ along $\text{span}(\Theta_0)$ and its orthogonal complement:

$$\Theta(t) = a^*(t)\Theta_0 + R(t), \quad \text{where} \quad a^*(t) = \arg\min_{a\in\mathbb{R}} ||\Theta(t) - a\Theta_0||.$$

By the orthogonality condition of least squares, we have:

$$a^*(t) = \frac{\langle\Theta(t), \Theta_0\rangle}{||\Theta_0||^2} \quad \text{and} \quad R(t) = \Theta(t) - a^*(t)\Theta_0.$$

From the definition of Frobenius cosine similarity, we can obtain the similarity $S(t)$ between $\Theta(t)$ and $\Theta_0$:

$$S(t) := \frac{\langle\Theta(t), \Theta_0\rangle}{||\Theta(t)||\,||\Theta_0||},$$

and by the Pythagorean theorem:

$$||\Theta(t)||^2 = ||a^*(t)\Theta_0||^2 + ||R(t)||^2 = \frac{\langle\Theta(t), \Theta_0\rangle^2}{||\Theta_0||^2} + ||R(t)||^2,$$

we can derive the residual norm:

$$||R(t)||^2 = ||\Theta(t)||^2 - \frac{\langle\Theta(t), \Theta_0\rangle^2}{||\Theta_0||^2} = ||\Theta(t)||^2(1 - S(t)^2),$$

so that

$$||R(t)|| = ||\Theta(t)||\sqrt{1 - S(t)^2}.$$

Now we define the scaled equivalent kernel $\Theta_{\text{eq}}(t)$ as:

$$\Theta_{\text{eq}}(t) := \frac{1}{a^*(t)}\Theta(t) = \Theta_0 + E(t), \quad \text{where} \quad E(t) = \frac{R(t)}{a^*(t)}.$$

Using the identity:

$$a^*(t) = \frac{||\Theta(t)||}{||\Theta_0||}S(t),$$

we bound the error term:

$$||E(t)|| = \frac{||R(t)||}{a^*(t)} = ||\Theta_0||\frac{\sqrt{1 - S(t)^2}}{S(t)}.$$

Since $S(t) \geq 1 - \epsilon$ and $S(t)^2 \geq 1 - 2\epsilon + \epsilon^2$, it follows that:

$$||E(t)|| \leq ||\Theta_0||\frac{\sqrt{2\epsilon - \epsilon^2}}{1 - \epsilon} \leq ||\Theta_0||\frac{\sqrt{2\epsilon}}{1 - \epsilon}.$$

Based on the expansion of $\Theta(t)$, the gradient flow can be transformed as:

$$\dot{f}(t) = -\eta\Theta(t)\gamma(t) = -\eta a^*(t)\Theta_0\gamma(t) - \eta R(t)\gamma(t).$$

Here we introduce the time reparameterization from $t$ to $u$:

$$u(t) = \int_0^t a^*(\tau)d\tau \quad \Rightarrow \quad \frac{du}{dt} = a^*(t).$$

Then, by the chain rule:

$$\dot{f}(u) = \frac{df/dt}{du/dt} = -\eta\Theta_0\gamma(t(u)) - \eta\frac{R(t(u))}{a^*(t(u))}\gamma(t(u)) = -\eta\Theta_0\gamma(t(u)) + \Delta(u),$$

where

$$\Delta(u) = -\eta\frac{R(t(u))}{a^*(t(u))}\gamma(t(u)).$$

Using the spectral norm product bound, $\|E(t(u))\,\gamma(t(u))\| \leq \|E(t(u))\|_{\mathrm{op}}\|\gamma(t(u))\|$, and the operator norm bound $\|\cdot\|_{\mathrm{op}} \leq \|\cdot\|$, we obtain

$$\|\Delta(u)\| \leq \eta\,\|E(t(u))\|_{\mathrm{op}}\,\|\gamma(t(u))\| \leq \eta\,\|E(t(u))\|\,\|\gamma(t(u))\| \leq \eta\,\|\Theta_0\|\,\frac{\sqrt{2\epsilon}}{1-\epsilon}\,\|\gamma(t(u))\|,$$

where $\|\cdot\|$ denotes the Frobenius norm and $\|\cdot\|_{\mathrm{op}}$ is the spectral norm.

Thus, under the time variable $u$, the dynamics are equivalent to those of a fixed kernel $\Theta_0$ up to a perturbation $\Delta$ bounded as above, which completes the proof. □

## D. Derivation of Jacobian-free NTK Approximation

*Proof.* The derivation of Equation 5 is formulated as below:

$$
\begin{aligned}
\widetilde{\Theta}(\mathbf{x}, \mathbf{x}'; \theta_t) &= \left\langle \nabla_{\theta_t}\sum_{k=1}^{D} f_k(\mathbf{x};\theta_t), \nabla_{\theta_t}\sum_{k=1}^{D} f_k(\mathbf{x}',\theta_t) \right\rangle, \\
&= \left\langle \sum_{k=1}^{D}\nabla_{\theta_t} f_k(\mathbf{x};\theta_t), \sum_{k=1}^{D}\nabla_{\theta_t} f_k(\mathbf{x}',\theta_t) \right\rangle \\
&= \sum_{k=1}^{D}\sum_{m=1}^{D}\langle\nabla_{\theta_t} f_k(\mathbf{x};\theta_t), \nabla_{\theta_t} f_m(\mathbf{x}',\theta_t)\rangle \\
&= \sum_{k=m=1}^{D}\langle\nabla_{\theta_t} f_k(\mathbf{x};\theta_t), \nabla_{\theta_t} f_m(\mathbf{x}',\theta_t)\rangle + \sum_{k=1}^{D}\sum_{m=1,m\neq k}^{D}\langle\nabla_{\theta_t} f_k(\mathbf{x};\theta_t), \nabla_{\theta_t} f_m(\mathbf{x}',\theta_t)\rangle \\
&= \sum_{k=1}^{D}\langle\nabla_{\theta_t} f_k(\mathbf{x};\theta_t), \nabla_{\theta_t} f_k(\mathbf{x}';\theta_t)\rangle + \sum_{k=1}^{D}\sum_{m=1,m\neq k}^{D}\langle\nabla_{\theta_t} f_k(\mathbf{x};\theta_t), \nabla_{\theta_t} f_m(\mathbf{x}',\theta_t)\rangle.
\end{aligned}
$$

□

## E. Theoretical Justification of the Jacobian-Free NTK Approximation

In this part, we provide a theoretical justification for why cross-output gradient terms in the calculation of NTK can be neglected for LLMs. We first recall the pseudo-NTK (pNTK) construction of Mohamadi & Sutherland (2022), who show that the eNTK is asymptotically proportional to the identity across outputs in wide randomly initialised networks. We then show that our Jacobian-free NTK approximation coincides with their pNTK up to a constant factor, and extend the argument to pretrained, finite-width LLMs under mild high-dimensional assumptions.

### E.1. Pseudo-NTK

Consider a general multi-output network $f_\theta(x) \in \mathbb{R}^D$ with parameters $\theta \in \mathbb{R}^P$ and output dimension $D$, let

$$J_\theta(f_\theta(\mathbf{x})) \in \mathbb{R}^{D \times P}$$

denote the Jacobian of the outputs with respect to all parameters. The NTK between two inputs $\mathbf{x}, \mathbf{x}'$ is denoted as

$$\Theta(\mathbf{x}, \mathbf{x}'; \theta) = J_\theta(f(\mathbf{x}; \theta)) \, J_\theta(f(\mathbf{x}'; \theta))^\top \in \mathbb{R}^{D \times D}. \tag{7}$$

[Mohamadi & Sutherland (2022)](#) introduce the pseudo-NTK (pNTK) as the scalar NTK of a single-output network obtained by averaging all outputs:

$$\widehat{\Theta}(\mathbf{x}, \mathbf{x}'; \theta) = \left\langle \nabla_\theta \frac{1}{\sqrt{D}} \sum_{k=1}^D f_k(\mathbf{x}; \theta), \ \nabla_\theta \frac{1}{\sqrt{D}} \sum_{k=1}^D f_k(\mathbf{x}'; \theta) \right\rangle, \tag{8}$$

where $f_k(\mathbf{x}; \theta)$ denotes the $k$-th output coordinate.

For fully connected networks with ReLU activations and width $n$ in each hidden layer, they show that at random initialization and as $n \to \infty$, the pNTK approximates the full eNTK in Frobenius norm:

$$\frac{\left\| \Theta^{(L)}(\mathbf{x}, \mathbf{x}'; \theta) - \widehat{\Theta}^{(L)}(\mathbf{x}, \mathbf{x}'; \theta) \, I_D \right\|_F}{\left\| \Theta^{(L)}(\mathbf{x}, \mathbf{x}'; \theta) \right\|_F} = O_{\mathbb{P}}(n^{-1/2}), \qquad n \to \infty, \tag{9}$$

where $L$ is the number of layers and $I_D$ is the $D \times D$ identity. Equivalently, the eNTK matrix becomes asymptotically proportional to $I_D$. Moreover, they show that the ratio of "information" in off-diagonal versus diagonal entries of the eNTK matrix decays as $O(n^{-1/2})$, reflecting the fact that gradients associated with different outputs become nearly orthogonal in parameter space.

Our Jacobian-free NTK approximation is exactly a rescaled version of the pNTK in Equation (8), but we apply it to pretrained finite-width LLMs instead of randomly initialized fully-connected networks. We provide a detailed explanation of the feasibility and make this connection precise, extending it to our setting under high-dimensional assumptions.

### E.2. Decomposition of Gradients

For an input $\mathbf{x}$, let $h(\mathbf{x}; \psi) \in \mathbb{R}^d$ denote the final hidden representation produced by the backbone, parametrized by $\psi$. Let $W \in \mathbb{R}^{D \times d}$ be the LM head with $k$-th row $w_k^\top$. The output logits are denoted as

$$f(\mathbf{x}; \theta) = W \, h(\mathbf{x}; \psi) \in \mathbb{R}^D, \qquad \theta = (W, \psi). \tag{10}$$

Correspondingly, the $k$-th output dimension is $f_k(\mathbf{x}; \theta) = w_k^\top h(\mathbf{x}; \psi)$.

For each output coordinate $k$, the gradient with respect to $P$ parameters is formulated as

$$g_k(\mathbf{x}) = \nabla_\theta f_k(\mathbf{x}; \theta) \in \mathbb{R}^P. \tag{11}$$

Using the product structure $\theta = (W, \psi)$, we decompose the gradient into two parts as

$$g_k(x) = \left[ \frac{\partial f_k}{\partial W}(\mathbf{x}; \theta), \ \frac{\partial f_k}{\partial \psi}(\mathbf{x}; \theta) \right], \tag{12}$$

where each of them can be formulated as

$$\frac{\partial f_k}{\partial W}(\mathbf{x}; \theta) = e_k \otimes h(\mathbf{x}; \psi), \tag{13}$$

$$\frac{\partial f_k}{\partial \psi}(\mathbf{x}; \theta) = J_\psi(\mathbf{x})^\top w_k. \tag{14}$$

$e_k$ is the $k$-th standard basis vector in $\mathbb{R}^D$, $J_\psi(\mathbf{x}) = \frac{\partial h(\mathbf{x}; \psi)}{\partial \psi} \in \mathbb{R}^{d \times P_\psi}$ is the backbone Jacobian, and $\otimes$ denotes the Kronecker product. Thus

$$g_k(\mathbf{x}; \theta) = \left[ e_k \otimes h(\mathbf{x}; \psi); \ J_\psi(\mathbf{x})^\top w_k \right]. \tag{15}$$

Based on this, we reformulate the standard NTK $\Theta$ and Jacobian-free NTK approximated NTK $\widetilde{\Theta}$ as

$$\Theta(\mathbf{x}, \mathbf{x}'; \theta) = \sum_{k=1}^{D} \left\langle g_k(\mathbf{x}; \theta), g_k(\mathbf{x}'; \theta) \right\rangle. \tag{16}$$

and

$$\widetilde{\Theta}(\mathbf{x}, \mathbf{x}'; \theta) = \left\langle \nabla_\theta \sum_{k=1}^{D} f_k(\mathbf{x}; \theta), \ \nabla_\theta \sum_{k=1}^{D} f_k(\mathbf{x}'; \theta) \right\rangle \tag{17}$$

$$= \left\langle \sum_{k=1}^{D} g_k(\mathbf{x}), \ \sum_{m=1}^{D} g_m(\mathbf{x}') \right\rangle \tag{18}$$

$$= \sum_{k=1}^{D} \sum_{m=1}^{D} \left\langle g_k(\mathbf{x}), g_m(\mathbf{x}') \right\rangle. \tag{19}$$

### E.3. Decomposition of Cross-output Terms

Based on Equation (15), we decompose cross-output terms as

$$\left\langle g_k(\mathbf{x}; \theta), g_m(\mathbf{x}'; \theta) \right\rangle = \left\langle e_k \otimes h(\mathbf{x}), e_m \otimes h(\mathbf{x}') \right\rangle + \left\langle J_\psi(\mathbf{x})^\top w_k, J_\psi(\mathbf{x}')^\top w_m \right\rangle$$

$$= \underbrace{\delta_{km} \left\langle h(\mathbf{x}), h(\mathbf{x}') \right\rangle}_{\text{LM head}} + \underbrace{w_k^\top B(\mathbf{x}, \mathbf{x}') \, w_m}_{\text{backbone}}, \tag{20}$$

where we $B(\mathbf{x}, \mathbf{x}')$ is defined as the backbone Jacobian Gram-type matrix:

$$B(\mathbf{x}, \mathbf{x}') = J_\psi(\mathbf{x}) \, J_\psi(\mathbf{x}')^\top \in \mathbb{R}^{d \times d}. \tag{21}$$

For $\mathbf{x} = \mathbf{x}'$, $B(\mathbf{x}, \mathbf{x})$ is a standard Jacobian Gram matrix and is symmetric positive semidefinite. For $\mathbf{x} \neq \mathbf{x}'$, $B(\mathbf{x}, \mathbf{x}')$ need not be symmetric. For notational simplicity, we keep the notation $B(\mathbf{x}, \mathbf{x}')$.

Considering Equation (20) for $k \neq m$, we have

$$\left\langle g_k(\mathbf{x}), g_m(\mathbf{x}') \right\rangle = w_k^\top B(\mathbf{x}, \mathbf{x}') \, w_m, \tag{22}$$

and for $k = m$, it is

$$\left\langle g_k(\mathbf{x}), g_k(\mathbf{x}') \right\rangle = \left\langle h(\mathbf{x}), h(\mathbf{x}') \right\rangle + w_k^\top B(\mathbf{x}, \mathbf{x}') \, w_k. \tag{23}$$

We sum Equation (23) over $k$ to reformulate Equation (16) into

$$\Theta(\mathbf{x}, \mathbf{x}'; \theta) = \sum_{k=1}^{D} \left\langle g_k(\mathbf{x}), g_k(\mathbf{x}') \right\rangle$$

$$= D \left\langle h(\mathbf{x}), h(\mathbf{x}') \right\rangle + \sum_{k=1}^{D} w_k^\top B(\mathbf{x}, \mathbf{x}') \, w_k$$

$$= D \left\langle h(\mathbf{x}), h(\mathbf{x}') \right\rangle + \operatorname{Tr}\!\left( W B(\mathbf{x}, \mathbf{x}') W^\top \right), \tag{24}$$

and expand Equation (19) into

$$\widetilde{\Theta}(\mathbf{x}, \mathbf{x}'; \theta) = \sum_{k=1}^{D} \sum_{m=1}^{D} \left\langle g_k(\mathbf{x}), g_m(\mathbf{x}') \right\rangle$$

$$= D \left\langle h(\mathbf{x}), h(\mathbf{x}') \right\rangle + \mathbf{1}^\top W B(\mathbf{x}, \mathbf{x}') W^\top \mathbf{1}, \tag{25}$$

where $\mathbf{1} \in \mathbb{R}^D$ is the all-ones vector. Their difference is exactly the sum of cross-output backbone terms:

$$\Delta(\mathbf{x}, \mathbf{x}') = \widetilde{\Theta}(\mathbf{x}, \mathbf{x}'; \theta) - \Theta(\mathbf{x}, \mathbf{x}'; \theta) = \sum_{k \neq m} w_k^\top B(\mathbf{x}, \mathbf{x}') \, w_m. \tag{26}$$

Note that the LM head contribution $D\langle h(\mathbf{x}), h(\mathbf{x}')\rangle$ cancels in the difference $\widetilde{\Theta} - \Theta$, so only the backbone Jacobian enters $\Delta(\mathbf{x}, \mathbf{x}')$. In what follows we show that, under mild high-dimensional assumptions on $W$ and $B(\mathbf{x}, x')$, the magnitude of $\Delta(\mathbf{x}, \mathbf{x}')$ is small relative to $\Theta(\mathbf{x}, \mathbf{x}'; \theta)$.

### E.4. High-dimensional assumptions

We now state the two assumptions that drive our analysis. They are standard in the theory of wide networks, and empirically consistent with the behaviour we observe for pretrained LLMs.

**Assumption E.1** (Isotropic LM head). Conditionally on the backbone parameters $\psi$, the rows of the readout matrix $W \in \mathbb{R}^{D \times d}$ are independent, mean-zero, isotropic random vectors in $\mathbb{R}^d$:

$$w_k \sim \mathcal{N}\big(0, (\sigma_w^2/d)\, I_d\big), \qquad k = 1, \ldots, D, \tag{27}$$

independently.

**Justification of Assumption E.1.** The isotropic LM-head model is standard at random initialization in infinite-width NTK analyses (Jacot et al., 2018a) and in the Tensor Programs framework (Yang, 2020a;b), where weights at each layer are sampled as i.i.d. mean-zero vectors with covariance proportional to the identity. In supervised classification, Neural Collapse results (Papyan et al., 2020; Zhu et al., 2021) further show that, when training drives the empirical risk near zero, last-layer classifier weights converge to a simplex equiangular tight frame. Geometrically, this means that, in the feature subspace, the classifier rows become approximately equinorm and pairwise equiangular, i.e., close to an isotropic configuration.

Crucially, token-level language modeling is itself a multi-class classification problem at each position, with the vocabulary playing the role of class labels and the LM head acting as a linear classifier on the contextual representation $h(x; \psi)$. Recent work on Linguistic Collapse (Wu & Papyan, 2024) demonstrates that Neural-Collapse-like geometries also emerge in causal language models, despite strongly imbalanced vocabularies and the number of tokens (classes) exceeding the hidden dimension. It means that as model size and training scale increase, the LM-head rows and certain token-dependent feature prototypes become increasingly equinorm, equiangular, and aligned. These findings suggest that, in high dimensions, the LM head of a pretrained LLM is well-approximated by an ensemble of nearly isotropic row vectors in the relevant feature subspace.

In this work, we therefore model the LM head $W$ as having independent, mean-zero, approximately isotropic rows as in Assumption A1. This is not intended as an exact statistical description of any specific checkpoint, but as a high-dimensional regularity hypothesis consistent with both the standard NTK initialization theory and empirical observations of Neural-Collapse-like behaviour in language models (Wu & Papyan, 2024).

**Assumption E.2** (Approximate isotropy of the backbone Jacobian Gram). For any given input pair $(\mathbf{x}, \mathbf{x}')$, the backbone Jacobian Gram-type matrix admits an approximate decomposition

$$B(\mathbf{x}, \mathbf{x}') \approx \beta(\mathbf{x}, \mathbf{x}')\, I_d + U(\mathbf{x}, \mathbf{x}')\, \Lambda(\mathbf{x}, \mathbf{x}')\, U(\mathbf{x}, \mathbf{x}')^\top, \tag{28}$$

where $\beta(\mathbf{x}, x\mathbf{x}) > 0$, $U(\mathbf{x}, \mathbf{x}') \in \mathbb{R}^{d \times r}$ has orthonormal columns with $r \ll d$, and $\Lambda(\mathbf{x}, \mathbf{x}') \in \mathbb{R}^{r \times r}$ is diagonal with uniformly bounded entries.[1]

---

[1] Formally, the decomposition is best viewed as describing the spectrum of the symmetric part $\frac{1}{2}(B + B^\top)$, and we drop the symmetrization from the notation for readability.

**Justification of Assumption E.2.** At random initialization and in wide networks, there are analyses of Jacobians that show that their singular values concentrate in a bounded interval, so the Jacobian behaves approximately like a scaled orthogonal transform (Pennington et al., 2017; Dadoun et al., 2025). Empirical studies (Wang et al., 2016) further suggest a spectrum with a few large singular values and a bulk of smaller ones, well approximated by a low-rank perturbation of an approximately isotropic bulk.

In the context of LLMs, there are several independent spectral studies that strongly support such a 'bulk + spikes' picture. First, plenty of work on activation geometry in LLMs shows that contextual representations are dominated by a small number of 'outlier dimensions' with very large variance, while the remaining coordinates form a comparatively flat bulk (Hämmerl et al., 2023; Rudman et al., 2023; Rudman & Eickhoff, 2023). From the perspective of covariance, this is precisely a low-rank anisotropic component on top of an approximately isotropic background in the hidden-state covariance. Second, recent Hessian-spectral analyses have reported an analogous structure: a small number of large eigenvalues together with a heavy-tailed or nearly-zero bulk, and a robust low-dimensional subspace that captures most of the curvature (QianYuan et al., 2025; Granziol, 2025). Since the loss Hessian can be written schematically as a weighted sum of Jacobian Gram matrices, these observations strongly suggest that $J_\psi(\mathbf{x})J_\psi(\mathbf{x})^\top$ itself exhibits a dominant low-rank component plus an approximately isotropic bulk in high dimensions. Finally, Aubry et al. (2024) directly studies the singular value decompositions of residual block Jacobians in a range of pretrained LLMs and finds that the top singular vectors align across depth, indicating that a small set of preferred directions in representation space governs the dynamics while the remaining directions behave more uniformly.

Taken together, these results motivate modelling $B(\mathbf{x}, \mathbf{x}')$ as in Assumption E.2: a bulk term $\beta(\mathbf{x}, \mathbf{x}')I_d$ capturing the high-dimensional isotropic component and a low-rank term $U(\mathbf{x}, \mathbf{x}')\Lambda(\mathbf{x}, \mathbf{x}')U(\mathbf{x}, \mathbf{x}')^\top$ capturing a few task-specific directions. As with Assumption E.1, we do not claim this decomposition to hold exactly for any given checkpoint. Rather, we view it as a high-dimensional regularity hypothesis consistent with current empirical evidence for LLMs. Our bounds only depend on the existence of a bounded-rank anisotropic component and an approximately isotropic bulk, so moderate deviations from this idealized structure merely affect constant factors and not the asymptotic $d$-scaling.

### E.5. Error bound of cross-output NTK terms

We now quantify the effect of cross-output terms in $\Delta(\mathbf{x}, \mathbf{x}')$ under Assumptions E.1 and E.2. For clarity, we begin with the purely isotropic backbone model $B = \beta I_d$, which recovers the $O(d^{-1/2})$ as proved in Mohamadi & Sutherland (2022), and then treat the general low-rank case.

**Isotropic backbone** $(B = \beta I_d)$. Assume $B(\mathbf{x}, \mathbf{x}') = \beta(\mathbf{x}, \mathbf{x}')I_d$, we suppress $(\mathbf{x}, \mathbf{x}')$ for brevity to obtain $B = \beta I_d$, $\beta > 0$. Define $\langle g_k(\mathbf{x}), g_m(\mathbf{x}')\rangle$ as $X_{km}$, we have

$$X_{km} = w_k^\top B w_m = \beta \, w_k^\top w_m, \qquad X_{kk} = w_k^\top B w_k = \beta \, \|w_k\|^2. \tag{29}$$

For $k \neq m$, we have $\mathbb{E}[X_{km}] = 0$ and, using Assumption E.1,

$$\mathbb{E}[X_{km}^2] = \beta^2 \, \mathbb{E}\big[(w_k^\top w_m)^2\big]$$

$$= \beta^2 \sum_{i=1}^{d} \mathbb{E}[w_{k,i}^2] \, \mathbb{E}[w_{m,i}^2]$$

$$= \beta^2 \, d\Big(\frac{\sigma_w^2}{d}\Big)^2 = \beta^2 \, \frac{\sigma_w^4}{d}. \tag{30}$$

Thus

$$\mathrm{std}(X_{km}) = \sqrt{\mathbb{E}[X_{km}^2]} = \beta \, \frac{\sigma_w^2}{\sqrt{d}}. \tag{31}$$

On the other hand,

$$\mathbb{E}[X_{kk}] = \beta \, \mathbb{E}\|w_k\|^2 = \beta \, \frac{\sigma_w^2}{d} \, \mathrm{Tr}(I_d) = \beta \, \sigma_w^2. \tag{32}$$

Hence, a single cross-output term is smaller than a typical diagonal backbone term by a factor

$$\frac{\mathrm{std}(X_{km})}{\mathbb{E}[X_{kk}]} = d^{-1/2}. \tag{33}$$

Now consider the sum of all cross-output terms

$$\Delta = \sum_{k \neq m} X_{km}. \tag{34}$$

Under Assumption E.1, the family $\{X_{km}\}_{k \neq m}$ has zero mean and weak dependencies. A variance calculation yields

$$\mathrm{Var}(\Delta) \lesssim D(D-1)\,\mathrm{Var}(X_{km}) = D(D-1)\,\beta^2\,\frac{\sigma_w^4}{d}, \tag{35}$$

where $\lesssim$ hides constants arising from the covariance terms. Hence

$$\mathrm{std}(\Delta) \lesssim \beta\,\sigma_w^2\,\sqrt{\frac{D(D-1)}{d}}. \tag{36}$$

The backbone contribution to the standard NTK can be represented as

$$S = \sum_{k=1}^{D} X_{kk}, \qquad \mathbb{E}[S] = D\,\beta\,\sigma_w^2. \tag{37}$$

Combining yields

$$\frac{\mathrm{std}(\Delta)}{\mathbb{E}[S]} \lesssim \sqrt{\frac{D-1}{D}} \cdot d^{-1/2} = O(d^{-1/2}), \qquad d \to \infty. \tag{38}$$

Note that the LM head term $D\langle h(\mathbf{x}), h(\mathbf{x}')\rangle$ is identical in $\Theta$ and $\widetilde{\Theta}$, so it does not contribute to $\Delta$. It does appear in the denominator $\Theta(\mathbf{x}, \mathbf{x}')$, and thus can only reduce the relative error $|\Delta|/\Theta$.

**Backbone with low-rank structure** ($B = \beta I_d + U\Lambda U^\top$). We now treat the general case of Assumption E.2, where we also suppress $(\mathbf{x}, \mathbf{x}')$ for brevity as

$$B = \beta I_d + U\Lambda U^\top, \tag{39}$$

and decompose

$$w_k^\top B w_m = \underbrace{\beta\,w_k^\top w_m}_{X_{km}^{\mathrm{bulk}}} + \underbrace{(U^\top w_k)^\top \Lambda (U^\top w_m)}_{X_{km}^{\mathrm{low}}}. \tag{40}$$

The bulk terms $X_{km}^{\mathrm{bulk}}$ are exactly as discussed above and yield an $O_{\mathbb{P}}(d^{-1/2})$ relative error. It remains to bound the low-rank contribution.

Let $u_k = U^\top w_k \in \mathbb{R}^r$. Since $U$ has orthonormal columns and $w_k$ are isotropic Gaussian under Assumption E.1, we have

$$u_k \sim \mathcal{N}\big(0, (\sigma_w^2/d)I_r\big), \qquad k = 1, \ldots, D, \tag{41}$$

independently. Then

$$X_{km}^{\mathrm{low}} = u_k^\top \Lambda u_m, \qquad k \neq m. \tag{42}$$

For $k \neq m$, $\mathbb{E}[X_{km}^{\mathrm{low}}] = 0$ and

$$\begin{aligned}
\mathbb{E}\big[(X_{km}^{\mathrm{low}})^2\big] &= \sum_{i=1}^{r} \lambda_i^2\,\mathbb{E}[u_{k,i}^2]\,\mathbb{E}[u_{m,i}^2] \\
&= \sum_{i=1}^{r} \lambda_i^2 \Big(\frac{\sigma_w^2}{d}\Big)^2 = \frac{\sigma_w^4}{d^2}\,\|\Lambda\|_F^2,
\end{aligned} \tag{43}$$

where $\lambda_i$ are the diagonal entries of $\Lambda$ and $\|\Lambda\|_F$ is its Frobenius norm. Thus

$$\mathrm{std}(X_{km}^{\mathrm{low}}) = \frac{\sigma_w^2}{d}\,\|\Lambda\|_F \leq \frac{\sigma_w^2}{d}\,\sqrt{r}\,\|\Lambda\|_2, \tag{44}$$

where $\|\Lambda\|_2$ is the spectral norm.

Similarly, the diagonal low-rank contribution is

$$Y_k^{\text{low}} = u_k^\top \Lambda u_k, \qquad \mathbb{E}[Y_k^{\text{low}}] = \frac{\sigma_w^2}{d} \, \text{Tr}(\Lambda). \tag{45}$$

For a single pair $(k, m)$, the ratio

$$\frac{\text{std}(X_{km}^{\text{low}})}{\mathbb{E}[Y_k^{\text{low}}]} = \frac{\|\Lambda\|_F}{\text{Tr}(\Lambda)} \leq \frac{\sqrt{r}\,\|\Lambda\|_2}{\text{Tr}(\Lambda)}$$

does not depend on $d$. Under the mild condition that $\text{Tr}(\Lambda)$ and $\sqrt{r}\|\Lambda\|_2$ are of the same order (no extreme anisotropy within the low-rank subspace), this ratio is $O(1)$. Thus low-rank cross-terms are not individually suppressed by $d^{-1/2}$. Instead, their high-dimensional smallness comes from the dominance of the isotropic bulk when we normalise by the total backbone NTK.

Therefore, we aggregate over all $(k, m)$ and combining bulk and low-rank parts and obtain

$$\frac{\left|\widetilde{\Theta}(\mathbf{x}, \mathbf{x}') - \Theta(\mathbf{x}, \mathbf{x}')\right|}{|\Theta(\mathbf{x}, \mathbf{x}')|} = O_{\mathbb{P}}\left(d^{-1/2} + \frac{\sqrt{r}}{d}\right), \qquad d \to \infty. \tag{46}$$

## F. Empirical NTK Regression

### F.1. Kernel ridge regression

Given a model $f(\cdot; \theta)$ with NTK $\Theta(\cdot, \cdot; \theta) \in \mathbb{R}^{n \times n}$ on a dataset $\mathcal{X} = \{\mathbf{x}_i, \mathbf{y}_i\}, i = 1 \ldots n$, we can perform standard kernel ridge regression (KRR) to predict the unseen test sample. For brevity, the model parameter $\theta_t$ is omitted in this part. The objective of KRR is to find a function $g$ within the Reproducing Kernel Hilbert Space (RKHS) (Berlinet & Thomas-Agnan, 2011) $\mathcal{H}_\Theta$ that minimizes a regularized squared error loss, which is formulated as

$$\min_{g \in \mathcal{H}_\Theta} \frac{1}{n} \sum_{i=1}^{n} (g(\mathbf{x}_i) - \mathbf{y}_i)^2 + \lambda \|g\|_{\mathcal{H}_\Theta}^2, \tag{47}$$

where $\lambda > 0$ is a user-defined regularization parameter. By the Representer Theorem, the optimal solution $g^*$ can be expressed as a linear combination of the kernel functions evaluated at the training data points:

$$g^*(\mathbf{x}) = \sum_{i=1}^{n} \alpha_i \Theta(\mathbf{x}, \mathbf{x}_i). \tag{48}$$

Substituting this representation into Equation (47) yields a convex optimization problem with respect to the coefficient vector $\alpha = [\alpha_1, \ldots, \alpha_n]^\top$. Therefore, Equation (47) can be rewritten as

$$\min_{\alpha} \frac{1}{n} \|\Theta(\cdot, \cdot)\alpha - \mathbf{y}\|^2 + \lambda \alpha^\top \Theta \alpha. \tag{49}$$

By setting the gradient with respect to $\alpha$ to zero, we can obtain a closed-form solution as

$$\alpha = (\Theta(\cdot, \cdot) + n\lambda I)^{-1} \mathbf{y}, \tag{50}$$

where $I \in \mathbb{R}^{n \times n}$ is the identity matrix. With the optimal coefficient vector $\alpha$ determined, we can obtain prediction $\hat{\mathbf{y}}_{\text{test}}$ for a new, unseen test sample $\mathbf{x}_{\text{test}}$ by applying the learned linear combination to the kernel values between the test sample and the entire training set, which is formulated as

$$\hat{\mathbf{y}}_{\text{test}} = \Theta_{\text{test}}(\mathbf{x}_{\text{test}}, \cdot)\alpha, \tag{51}$$

where $\Theta_{\text{test}}(\mathbf{x}_{\text{test}}, \cdot) \in \mathbb{R}^{1 \times n}$ is a vector containing the empirical NTK values between $\mathbf{x}_{\text{test}}$ and each of the training samples.

## F.2. Implementation

Due to the intricate nature of LLM outputs, which involve generating a step-by-step reasoning path followed by a final answer, directly applying a kernel regression model to this process is challenging. To address this, we simplify the problem by reframing it as a classification task. Specifically, for each task, we align the label sets of the training and test data to ensure class consistency. For instance, in the FPB task, we unify the original seven sentiment classes (strong negative, moderately negative, mildly negative, neutral, mildly positive, moderately positive, strong positive) from part of the training set into three broader categories: negative, neutral, and positive. This re-labeling allows us to align the classes with the test set, enabling a direct comparison of performance.

Our experiments are conducted by training the model on 1,000 domain samples and evaluating its performance on the corresponding test sets. We compare the classification accuracy of our regression-based approach against standard fine-tuning methods. The optimal regularization parameter $\lambda$ for our regression model is determined through a hyperparameter search over the range [1e-5, 1e-4, 3e-4, 1e-3, 3e-3, 1e-2, 1e-1]. It is important to note that this experimental setup could not be applied to the medical question-answering tasks (MedMCQA and MMLU-Med), as their answer options for individual instances are not consistent across the datasets. Consequently, direct label alignment for these tasks was not feasible. Therefore, we did not report results for these specific tasks in Table 3.

# G. Experimental Details

## G.1. Tasks

**MedMCQA.**   Medical Multiple-Choice Question Answering (MedMCQA) (Pal et al., 2022) is a benchmark for medical question answering, designed to address real-world medical entrance exam questions. It comprises 2.4K healthcare topics across 21 medical disciplines, with each question featuring four answer choices. This task necessitates a sophisticated integration of semantic comprehension, extensive factual recall, and advanced logical and causal reasoning. We train on UltraMedical (Zhang et al., 2024) and evaluate on the MedMCQA test set (4,183 samples).

**MMLU-Med.**   Massive Multitask Language Understanding (MMLU) (Hendrycks et al., 2021) serves as a multitask language understanding benchmark consisting of 57 tasks. In this paper, we exclusively adopt its medical-related subset (1,089 samples), MMLU-Med, which is sourced from various public and academic datasets, covering a wide range of medical subfields from clinical medicine to professional ethics. Distinct from MedMCQA, this task is designed to assess the generalization and robust knowledge of a pre-trained model rather than its fine-tuned performance on a narrow domain. We similarly leverage the UltraMedical (Zhang et al., 2024) corpus for training. More broadly, recent efforts have also begun to build domain-specific foundations in healthcare beyond text QA, such as body-sound diagnostics resources (Wang et al., 2024c; Zhao et al., 2024; 2025).

**FPB & TFNS.**   Financial PhraseBank (FPB) (Malo et al., 2014) and Twitter Financial News Sentiment (TFNS) (Magic, 2022) constitute benchmarks for financial sentiment analysis, with FPB providing sentence-level annotations and TFNS covering Twitter-based financial news. Models must classify sentiment into positive, neutral, or negative, requiring robust semantic understanding of context-dependent financial language and fine-grained classification across formal documents and informal social media. We employ their sentiment datasets for instruction tuning (Liu et al., 2023b) and evaluate on a random sample of 1,000 test instances.

**Headline**   (Sinha & Khandait, 2021) is a fine-grained classification benchmark for commodity market news headlines. It comprises 11,412 human-annotated gold-related headlines spanning from 2000 to 2019, requiring models to capture nuanced aspects including price direction, temporal context, and asset comparisons. Unlike sentiment analysis, Headline necessitates precise classification of price movements through contextual cues and distinction between historical and forward-looking trends, which enable the extraction of actionable insights for investors and policymakers. Considering the large quantity of the test set, we evaluate on a randomly sampled subset of 1,000 instances.

**ContractNLI**   (Koreeda & Manning, 2021) is a benchmark for evaluating a model's ability to perform document-level natural language inference (NLI) on contract texts to address the significant time and cost associated with manual contract review. The model is given a contract and a set of hypotheses, and for each hypothesis, it must classify whether it is entailed, contradictory, or neutral to the contract. This task is particularly challenging due to the unique linguistic properties of legal

documents and the complexity of document-level reasoning. Similarly, we adopted a randomly sampled test set of 1,000 data points.

**PrivacyQA** (Ravichander et al., 2019) assesses a model's ability to answer questions based on a privacy policy document. This task is framed as a question-answering problem where a model is given a question about a privacy policy and must provide a concise, factual answer. Building on this, it requires the model to not only read and understand a lengthy, complex legal document but also to locate specific information and synthesize a correct response. We use a random sample of 1,000 examples for held-out evaluation.

**RSDD.** Reddit Self-reported Depression Diagnosis (RSDD) (Yates et al., 2017) task is a benchmark for evaluating a model's capability to identify users with depression from their online forum language alone. This task is framed as a user-level binary classification problem, where a system must determine if a user, based on the corresponding posting history, has a self-reported depression diagnosis or belongs to a matched control group. Beyond simple keyword matching, this task requires the model to capture nuanced linguistic and socio-linguistic patterns associated with mental health. As the dataset lacks predefined splits, we evaluate on a random sample of 1,000 instances and use the remainder for training.

### G.2. Baselines

We now detail the implementation of the data selection baselines. For random selection, a single auxiliary dataset is sampled once from the full candidate pool and shared across all tasks to ensure consistency. For embedding-based and gradient-based selection, we average the hidden states of the last layer or loss gradients, and each candidate is scored by its average similarity to all domain samples to select the top $N$ samples. For other task-specific selection approaches like DSIR[2] (Xie et al., 2023), we consider the domain dataset $\mathcal{D}$ as the high-quality targets and identify similar data from the candidate pool $\mathcal{C}$. For LESS[3] (Xia et al., 2024), due to the large scale of candidate pool $\mathcal{C}$, we perform warm-up training on 0.1% ($\approx$2K samples) of the candidate data. Following the original setup, Adam gradients are used for candidates and SGD gradients for domain samples. For TSDS[4] (Liu et al., 2024), we adopt the recommended hyperparameters: scaling constant $C = 5.0$, diversity-alignment trade-off coefficient $\alpha = 0.5$, and prefetching/kernel density estimation neighborhood sizes $K = 5000$ and $K = 1000$, respectively.

### G.3. Training

We report the LoRA-based instruction tuning configuration in Table 5. The same training setup is consistently applied across all data combinations generated via distinct selection strategies to ensure fair comparison. This LoRA configuration is also shared with the NTK selection stage.

*Table 5.* Training details of LoRA-based instruction tuning.

| Parameter Name | Value |
| --- | --- |
| LoRA $\alpha$ | 32 |
| LoRA $r$ | 16 |
| LoRA dropout | 0.05 |
| LoRA target modules | q_proj, k_proj, v_proj, o_proj, up_proj, down_proj, gate_proj |
| Epoch | 3 |
| Warm-up ratio | 0.03 |
| Weight decay | 0.0 |
| Learning rate | 5e-4 |
| Learning rate schedule | Cosine |
| Max sequence length | 3072 |
| Batch size per device | 8 |
| Gradient accumulation steps | 8 |
| Platform | 4 NVIDIA A100 Tensor Core GPU |

---

[2] https://github.com/p-lambda/dsir
[3] https://github.com/princeton-nlp/LESS
[4] https://github.com/ZifanL/TSDS

*Table 6.* Cosine similarity of NTK matrices between Epoch 0 (the initial state) and Epoch 1∼10, which are calculated by LLAMA3-8B-INSTRUCT and QWEN3-8B for financial, medical, legal, and psychological domains with 3 independent runs.

| Domain | Epoch 1 | Epoch 2 | Epoch 3 | Epoch 4 | Epoch 5 | Epoch 6 | Epoch 7 | Epoch 8 | Epoch 9 | Epoch 10 |
|---|---|---|---|---|---|---|---|---|---|---|
| | | | | | LLAMA3-8B-INSTRUCT | | | | | |
| Financial sentiment analysis | 0.9793±0.0281 | 0.9992±0.0001 | 0.9991±0.0001 | 0.9992±0.0001 | 0.9992±0.0001 | 0.9992±0.0000 | 0.9992±0.0001 | 0.9991±0.0000 | 0.9991±0.0000 | 0.9991±0.0000 |
| Medical QA | 0.9991±0.0001 | 0.9984±0.0000 | 0.9982±0.0001 | 0.9975±0.0003 | 0.9965±0.0006 | 0.9954±0.0008 | 0.9945±0.0009 | 0.9940±0.0009 | 0.9938±0.0010 | 0.9938±0.0010 |
| Legal NLI | 0.9997±0.0000 | 0.9995±0.0001 | 0.9996±0.0001 | 0.9995±0.0000 | 0.9995±0.0000 | 0.9995±0.0000 | 0.9995±0.0000 | 0.9995±0.0001 | 0.9994±0.0001 | 0.9994±0.0001 |
| Depression diagnosis | 0.9996±0.0001 | 0.9996±0.0001 | 0.9989±0.0000 | 0.9990±0.0000 | 0.9990±0.0001 | 0.9987±0.0004 | 0.9985±0.0006 | 0.9984±0.0006 | 0.9983±0.0006 | 0.9983±0.0006 |
| | | | | | QWEN3-8B | | | | | |
| Financial sentiment analysis | 0.9991±0.0001 | 0.9985±0.0001 | 0.9982±0.0001 | 0.9981±0.0001 | 0.9979±0.0003 | 0.9977±0.0003 | 0.9976±0.0002 | 0.9974±0.0003 | 0.9974±0.0002 | 0.9973±0.0003 |
| Medical QA | 0.9992±0.0001 | 0.9990±0.0001 | 0.9986±0.0001 | 0.9980±0.0002 | 0.9970±0.0004 | 0.9954±0.0004 | 0.9937±0.0008 | 0.9917±0.0011 | 0.9922±0.0008 | 0.9789±0.0196 |
| Legal NLI | 0.9994±0.0000 | 0.9994±0.0001 | 0.9993±0.0001 | 0.9993±0.0001 | 0.9993±0.0001 | 0.9992±0.0001 | 0.9991±0.0000 | 0.9991±0.0001 | 0.9991±0.0001 | 0.9991±0.0001 |
| Depression diagnosis | 0.9995±0.0001 | 0.9988±0.0002 | 0.9977±0.0004 | 0.9978±0.0004 | 0.9982±0.0002 | 0.9981±0.0001 | 0.9980±0.0001 | 0.9978±0.0001 | 0.9978±0.0001 | 0.9978±0.0001 |

## G.4. Evaluation

For evaluation, we employ Chain-of-Thought (CoT) evaluation for each task. This approach first generates the reasoning process and subsequently produces the final answer by using the question concatenated with the historical reasoning path as input. To facilitate a more efficient evaluation, we leverage vLLM (Kwon et al., 2023) to accelerate the generation process. Additionally, drawing upon the observations of (Jiang et al., 2024) to enhance tolerance to data mismatches, we adopt the TAIA (Jiang et al., 2024) evaluation strategy for medical and legal tasks when using mixed training with auxiliary data.

# H. More Experimental Results

## H.1. Empirical Results of NTK-like Behavior

### H.1.1. NTK-LIKE BEHAVIOR IN MORE MODELS AND TASKS

To validate the NTK-like behavior in more tasks with multiple independent runs, we calculate the cosine similarity of the obtained NTK matrices derived by different epochs through the training, and report the mean values and standard deviation. As shown in Table 6, in most cases, the model exhibits a high similarity of over 0.99, which demonstrates that NTK-like behavior holds for diverse models and tasks during finetuning.

### H.1.2. NTK-LIKE BEHAVIOR WITHIN MORE FINETUNING STRATEGIES

Beyond the original experiments on LLAMA3-8B-INSTRUCT and QWEN-8B with LoRA, we add results on two additional architectures, DEEPSEEK-R1-1.5B (Guo et al., 2025) and Hermes3-8B (Quesnelle & Guang), and we evaluate three fine-tuning strategies for all models: LoRA, IA3 (Liu et al., 2022), and full-parameter fine-tuning. For each configuration, we compute the NTK matrix on a fixed input set with 100 samples from the financial sentiment analysis domain at initialization and after the N-th epoch, and report the cosine similarity between the two. As shown in Table 7, the similarities remain consistently high across all four models and three strategies (typically ≥0.99, and ≥0.985 even in the most challenging full fine-tuning cases). These results indicate that NTK-like stability is not specific to a particular model or to LoRA, but appears robust across different architectures and both parameter-efficient and full-parameter fine-tuning in the regimes we study. We will include these experiments and a brief discussion in the revised version as soon as possible.

### H.1.3. NTK-LIKE BEHAVIOR WITH VARIOUS TRAINING STRATEGIES

To probe the limits of this regime, we extend training to 50 epochs and vary the learning rate and batch size, which are set as 2e-4 and 32 per device as detailed in Appendix G.3. Table 8 shows the NTK cosine similarity under these settings. When the learning rate is set to an unrealistically large value (2e-2), training diverges, and the NTK becomes NaN. For all reasonable hyperparameters (e.g., lr ∈ {2e-4, 2e-3}, batch size ∈ {1, 32}), the NTK remains highly stable even up to 50 epochs, approximately from 0.998 to 0.999. Intuitively, for well-pretrained LLMs fine-tuned with moderate learning rates and standard parameter-efficient-finetuning (PEFT) or full finetuning recipes, the adaptation behaves as a relatively small perturbation of the pretrained parameters, so the NTK does not rotate dramatically.

*Table 7.* Cosine similarity of NTK matrices between Epoch 0 (the initial state) and Epoch 1∼10, which are calculated by LLAMA3-8B-INSTRUCT, QWEN3-8B, DEEPSEEK-R1-1.5B, HERMES3-8B on the financial sentiment analysis domain.

| Strategy | Epoch 1 | Epoch 2 | Epoch 3 | Epoch 4 | Epoch 5 | Epoch 6 | Epoch 7 | Epoch 8 | Epoch 9 | Epoch 10 |
|---|---|---|---|---|---|---|---|---|---|---|
| | | | | | LLAMA3-8B-INSTRUCT | | | | | |
| LoRA | 0.9594 | 0.9993 | 0.9992 | 0.9993 | 0.9993 | 0.9992 | 0.9992 | 0.9991 | 0.9991 | 0.9991 |
| IA3 | 1.0000 | 0.9997 | 0.9998 | 0.9982 | 0.9988 | 0.9988 | 0.9988 | 0.9988 | 0.9988 | 0.9988 |
| Full | 0.9971 | 0.9967 | 0.9970 | 0.9965 | 0.9963 | 0.9961 | 0.9962 | 0.9961 | 0.9961 | 0.9961 |
| | | | | | QWEN-8B | | | | | |
| LoRA | 0.9991 | 0.9986 | 0.9983 | 0.9982 | 0.9981 | 0.9979 | 0.9977 | 0.9976 | 0.9975 | 0.9975 |
| IA3 | 1.0000 | 1.0000 | 0.9998 | 1.0000 | 1.0000 | 1.0000 | 1.0000 | 1.0000 | 1.0000 | 1.0000 |
| Full | 0.9878 | 0.9879 | 0.9873 | 0.9865 | 0.9862 | 0.9861 | 0.9858 | 0.9855 | 0.9853 | 0.9853 |
| | | | | | DEEPSEEK-R1-1.5B | | | | | |
| LoRA | 0.9995 | 0.9992 | 0.9991 | 0.9991 | 0.9991 | 0.9990 | 0.9990 | 0.9990 | 0.9990 | 0.9989 |
| IA3 | 1.0000 | 1.0000 | 1.0000 | 1.0000 | 1.0000 | 0.9999 | 0.9999 | 0.9999 | 0.9999 | 0.9999 |
| Full | 0.9989 | 0.9983 | 0.9983 | 0.9982 | 0.9981 | 0.9981 | 0.9980 | 0.9980 | 0.9979 | 0.9979 |
| | | | | | HERMES3-8B | | | | | |
| LoRA | 0.9918 | 0.9923 | 0.9924 | 0.9923 | 0.9923 | 0.9921 | 0.9917 | 0.9917 | 0.9913 | 0.9915 |
| IA3 | 1.0000 | 0.9995 | 0.9985 | 0.9994 | 0.9993 | 0.9992 | 0.9990 | 0.9991 | 0.9992 | 0.9992 |
| Full | 0.9899 | 0.9895 | 0.9905 | 0.9897 | 0.9892 | 0.9888 | 0.9888 | 0.9883 | 0.9883 | 0.9883 |

*Table 8.* Cosine similarity of NTK matrices between Epoch 0 (the initial state) and Epoch 1∼50 with an interval of 5, which are calculated by LLAMA3-8B-INSTRUCT with various hyperparameter settings on the financial sentiment analysis domain.

| Lr | Batch size | Epoch 1 | Epoch 5 | Epoch 10 | N=15 | Epoch 20 | Epoch 25 | Epoch 30 | Epoch 35 | Epoch 40 | Epoch 45 | Epoch 50 |
|---|---|---|---|---|---|---|---|---|---|---|---|---|
| **2e-4** | 32 | 0.9997 | 0.9995 | 0.9984 | 0.9986 | 0.9988 | 0.9988 | 0.9988 | 0.9989 | 0.9989 | 0.9989 | 0.9989 |
| **2e-2** | 32 | NaN | NaN | NaN | NaN | NaN | NaN | NaN | NaN | NaN | NaN | NaN |
| **2e-3** | 32 | 0.9993 | 0.9991 | 0.9989 | 0.9985 | 0.9990 | 0.9991 | 0.9989 | 0.9991 | 0.9990 | 0.9983 | 0.9983 |
| **2e-4** | 1 | 0.9995 | 0.9987 | 0.9976 | 0.9977 | 0.9982 | 0.9984 | 0.9988 | 0.9987 | 0.9987 | 0.9987 | 0.9987 |

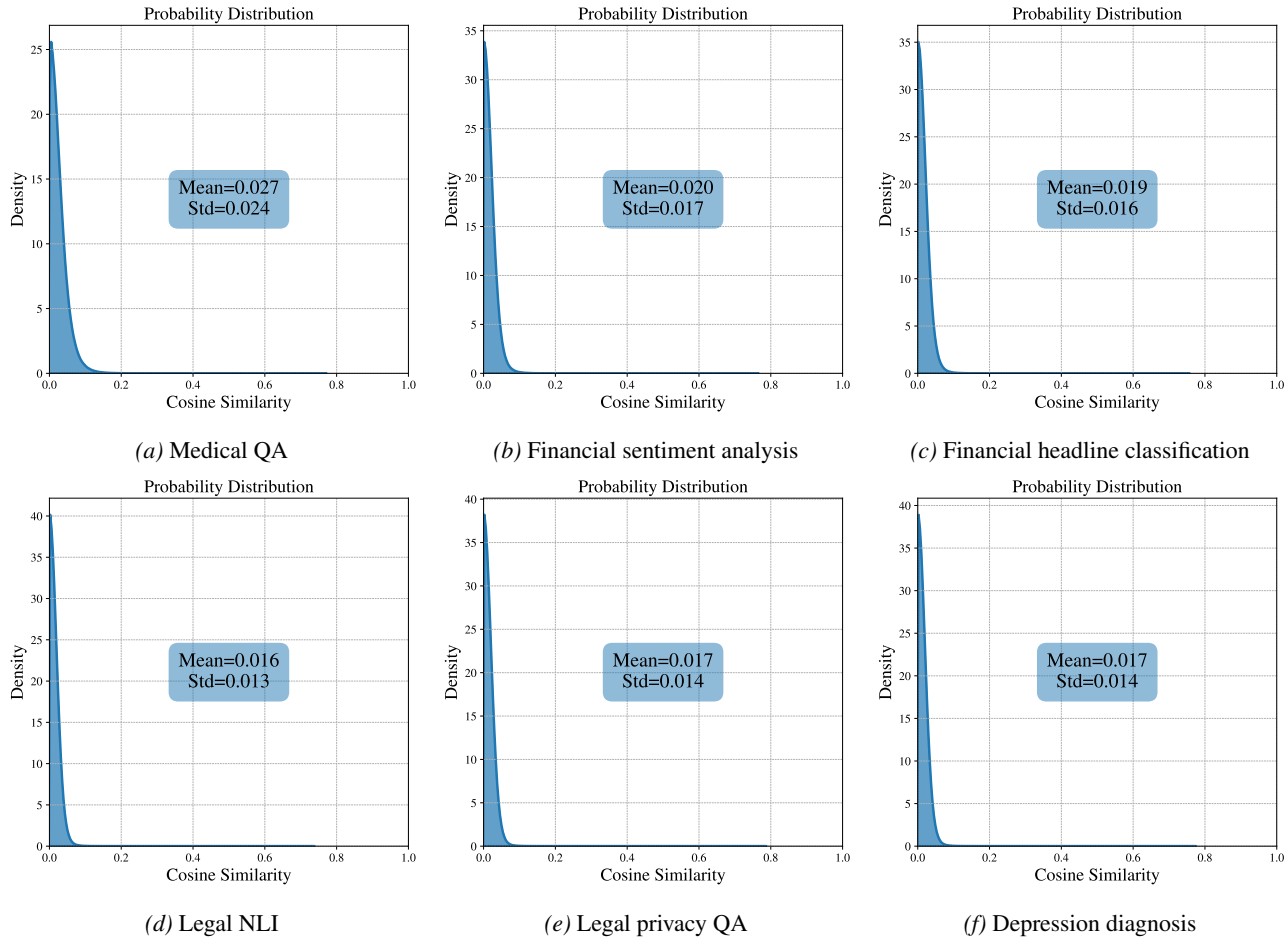

*Figure 6.* Gradient similarity of cross output of LLAMA3-8B-INSTRUCT on different tasks.

## H.2. Empirical Results of Jacobian-free Approximation

### H.2.1. GRADIENT SIMILARITY OF CROSS OUTPUT

This part visualizes how the gradients of cross channels correlate with each other. Considering an input $x$, the cross output gradient similarity between channel $k$ and $m$ is calculated as $\langle \nabla f_k(x;\theta), \nabla f_m(x;\theta) \rangle$. Figure 6 presents the gradient cosine similarity distribution of cross output on 100 randomly selected samples. Notably, most of the gradient similarities are below 0.1 with an average of around 0.02, supporting the practical validity of the near-orthogonality assumption underlying our Jacobian-free approximation as described in § 3.2.

### H.2.2. SIMILARITY BETWEEN THE APPROXIMATED NTK WITH THE STANDARD ONE

In this part, we compute the Spearman rank correlation between the Jacobian-free approximated NTK $\widetilde{\Theta}$ and the standard one $\Theta$. As shown in Table 9, the correlations are consistently above 0.95 across all models and tasks. This indicates that although our approximation introduces some numerical error, it preserves the relative ordering of NTK values very well, which is eactly what matters in our data selection application.

## H.3. Error of Random Projection

In this part, we quantify the error introduced by random projection by calculating the Spearman similarity of the calculated NTK matrices derived by using random projection or not. As Table 10 shows, random projection brings negligible error to the calculated NTK matrices, therefore enhancing accurate data selection.

*Table 9.* Spearman rank correlation between the Jacobian-free approximated NTK and the standard one on different model architectures and domains.

| Model | Financial Sentiment Analysis | Medical QA | Legal NLI | Depression Diagnosis |
|---|---|---|---|---|
| LLAMA3-8B-INSTRUCT | 0.9924 | 0.9788 | 0.9896 | 0.9989 |
| QWEN3-8B | 0.9837 | 0.9566 | 0.9726 | 0.9980 |
| DEEPSEEK-R1-1.5B | 0.9907 | 0.9861 | 0.9801 | 0.9977 |
| HERMES-8B | 0.9885 | 0.9586 | 0.9932 | 0.9576 |

*Table 10.* Spearman rank correlation of the NTK matrices by applying random projection or not on different model architectures and domains

| | Financial Sentiment Analysis | Medical QA | Legal NLI | Depression Diagnosis |
|---|---|---|---|---|
| LLAMA3-8B-INSTRUCT | 0.9970 | 0.9968 | 0.9975 | 1.0000 |
| QWEN3-8B | 0.9969 | 0.9951 | 0.9963 | 0.9997 |
| DEEPSEEK-R1-1.5B | 0.9957 | 0.9968 | 0.9892 | 0.9995 |
| HERMES3-8B | 0.9870 | 0.9955 | 0.9972 | 0.9963 |

## H.4. Optimization Steps vs. Accuracy

In this part, we present how the task performance evolves under different target domain data budgets and auxiliary data. Specifically, we set the domain data budget $|\mathcal{D}|$ as 100, 200, 1000, and 2000, respectively. For each configuration, we augment with auxiliary samples by NTK-Selector to increase the data pool and therefore optimization steps to be $2\times, 5\times, 10\times, 20\times$. All the training settings hold the default as described in Appendix G.3. As Figure 7 shows, more optimization steps with auxiliary data points improve task performance significantly, especially under extremely low domain data budget conditions (e.g. $|\mathcal{D}| = 100$).

## H.5. Compute vs. Accuracy

This section aims to provide practical guidance on how to choose hyperparameter settings based on the computational resources budget. Specifically, we investigate the impact of two important hyperparameters: pre-selection size $M = 2N, 4N, 16N$ and projection size $p = 1024, 2048, 4096, 8192$, with the selection budget $N = 9000$ by default. Figure 8 demonstrates the relationship between wall-clock run time in NTK selection stage and average task performance on LLAMA3-8B-INSTRUCT. Table 8 exhibits the storage consumption of NTK selection under various hyperparameter settings. Within the acceptable computational resources budget, it is recommended to adopt a higher pre-selection size and projection size.

## H.6. Compute-matched Comparison with Domain-only

To prove the improvement does not stem from more optimization steps, we perform a compute-matched study where we increase the number of training epochs for Domain-only on Llama3-8B-Instruct by a factor of 10, so that the total number of update steps (file × epoch) roughly matches that of NTK-Selector. The results are shown in Table 12. We observe that increasing the compute for Domain-only from 1k×3 to 1k×30 leads to only a marginal average improvement (68.7 → 69.5), and in several tasks (e.g., MedMCQA, MMLU-Med, Headline), performance actually drops, suggesting overfitting rather than genuine gains. In contrast, NTK-Selector remains clearly ahead even when Domain-only is given a comparable update

*Table 11.* Storage consumption of NTK selection stage with different hyperparameter settings: pre-selection size $M = 2N, 4N, 16N$, projection dimension $p = 1024, 2048, 4096, 8192$. All the gradients are stored in torch.float32 format.

| | $p = 1024$ | $p = 2048$ | $p = 4096$ | $p = 8192$ |
|---|---|---|---|---|
| $M = 2N$ | 0.07 GB | 0.14 GB | 0.27 GB | 0.55 GB |
| $M = 4N$ | 0.14 GB | 0.27 GB | 0.55 GB | 1.10 GB |
| $M = 16N$ | 0.55 GB | 1.10 GB | 2.20 GB | 4.39 GB |

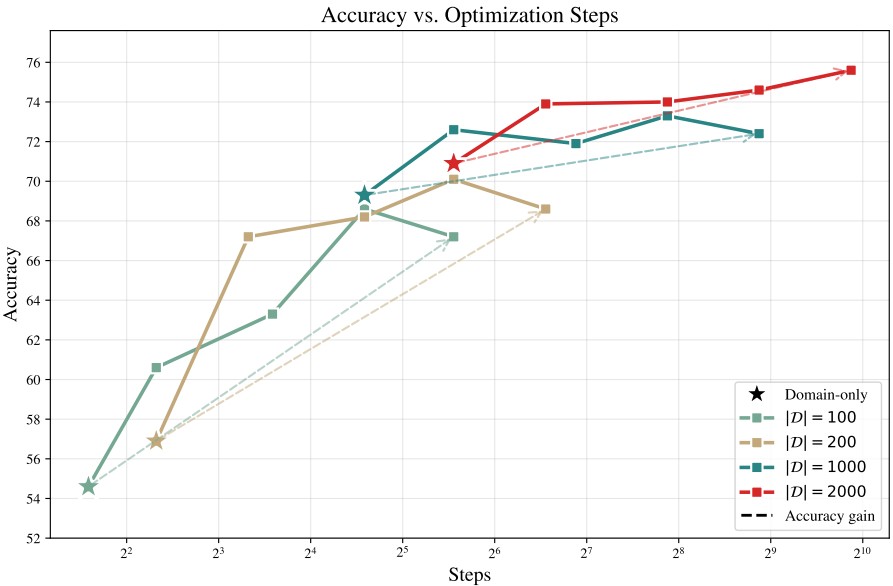

*Figure 7.* Optimization steps and accuracy of TFNS task on LLAMA3-8B-INSTRUCT under different target domain data budgets ($|\mathcal{D}| = 100, 200, 1000, 2000$) augmented with various numbers of auxiliary samples ($2\times, 5\times, 10\times, 20\times$). The starred point of each line denotes the Domain-only setting, and other dots represent the augmented setting with selected auxiliary data.

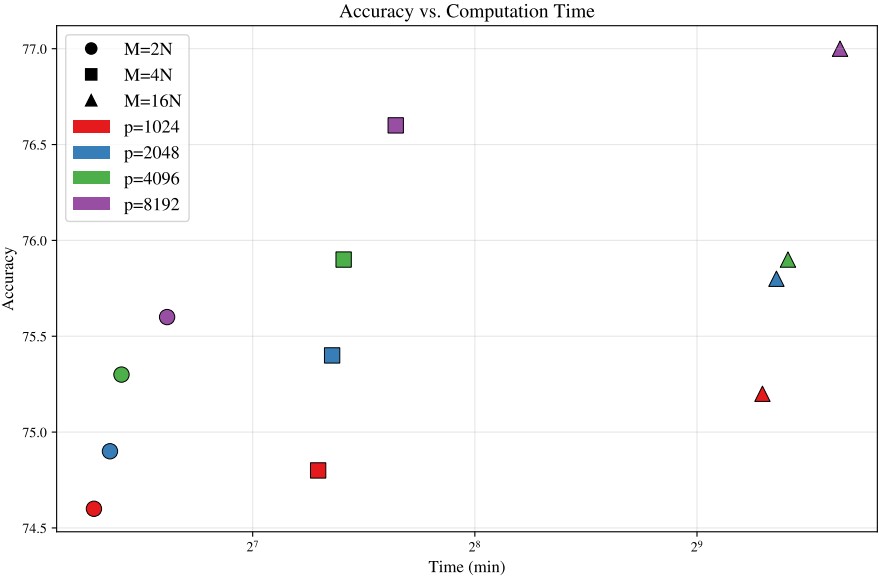

*Figure 8.* Time consumption and average accuracy on LLAMA3-8B-INSTRUCT with different hyperparameter settings: pre-selection size $M = 2N, 4N, 16N$, projection dimension $p = 1024, 2048, 4096, 8192$. The time consumption is computed for the NTK selection stage on one single A100 GPU.

*Table 12.* Compute-matched comparison between the domain-only setting and NTK-Selector.

| | Training budget (file * epoch) | MedMCQA | MMLU-Med | FPB | TFNS | Headline | ContractNLI | PrivacyQA | RSDD | Avg. |
|---|---|---|---|---|---|---|---|---|---|---|
| Base | - | 56.7 | 72.3 | 81.7 | 57.2 | 78.5 | 69.9 | 57.9 | 78.8 | 67.9 |
| Domain-only | 1k×3 | 56.5 | 71.8 | 80.1 | 69.3 | 82.4 | 41.1 | 63.6 | 85.1 | 68.7 |
| | 1k×30 | 54.8 | 69.3 | 81.2 | 71.7 | 80.3 | 43.4 | 69.1 | 86.2 | 69.5 |
| NTK-Selector | 10k×3 | **59.1** | **73.8** | **85.5** | **73.3** | **86.2** | **70.6** | **67.8** | **96.5** | **76.6** |

*Table 13.* Task performance by removing the warm-up stage in pre-selection on LLAMA3-8B-INSTRUCT.

| | MedMCQA | MMLU-Med | FPB | TFNS | Headline | ContractNLI | PrivacyQA | RSDD | Avg. |
|---|---|---|---|---|---|---|---|---|---|
| **NTK-Selector** | 59.1 | 73.8 | 85.5 | 73.3 | 86.2 | 70.6 | 67.8 | 96.5 | 76.6 |
| **- Warmup** | 58.7 | 73.2 | 85.8 | 72.7 | 85.6 | 71.6 | 66.9 | 95.8 | 76.3 |

budget. This indicates that our improvements are not simply due to higher compute, but to better data selection. Orthogonal to our data-centric gains, model-centric approaches have also improved deployment efficiency through retraining-free pruning and reconstruction (Wang et al., 2024b;a).

## H.7. Effect of Warm-up Stage in Pre-selection.

To analyze the effect of the warm-up stage in pre-selection, we perform an ablation on LLAMA3-8B-INSTRUCT to isolate the effect of the warm-up stage before NTK-based selection. Specifically, we remove the warm-up and compare against NTK-Selector. Table 13 shows that, even without warm-up, NTK-Selector still outperforms all baselines in average score, and is very close to the full version with warm-up. This indicates that the main performance gains come from the NTK-based selection itself, while the warm-up serves primarily as a modest enhancement to better exploit the limited target-domain data, rather than as a crucial extra-compute advantage.

## H.8. Documented Overlap

To investigate the documented overlap, we computed, for each text $x$ in CoT Collection and the test set for each task, the maximum 3-gram overlap $\max_i \text{sim}(x, y_i)$ with the test samples $\{y_1, y_2, \dots\}$ of that task, and then summarized the overlap distribution. As shown in Table 14, for all tasks, especially the legal and privacy benchmarks, almost all CoT Collection samples have very low overlap (<0.1) with any test example, and we find essentially no near-duplicates. These results suggest that, under a conservative 3-gram criterion, the candidate pool contains virtually no duplicated or near-duplicated test examples, and we do not observe evidence of leakage from CoT Collection into our evaluation sets.

## H.9. Why Augment with CoT.

Our decision to perform data selection on chain-of-thought (CoT) to annotate examples follows recent work showing that supervising models on intermediate reasoning steps, rather than only final answers, can substantially improve training effectiveness across tasks (Li et al., 2023; Hsieh et al., 2023). CoT labels provide a richer signal about the model's reasoning process, which is exactly what NTK-Selector aims to match. In preliminary experiments where the reference domain data contained only final answers (no CoT), we observed that selection tends to favor examples with extremely short or even

*Table 14.* Overlap proportion between Cot Collection and the test set of each task by calculating the 3-gram overlap.

| Overlap | MedMCQA | MMLU-Med | FPB | TFNS | Headline | ContractNLI | PrivacyQA | RSDD |
|---|---|---|---|---|---|---|---|---|
| **[0.00, 0.10)** | 99.96% | 99.99% | 99.87% | 100.00% | 100.00% | 100.00% | 100.00% | 100.00% |
| **[0.10, 0.20)** | 0.04% | 0.01% | 0.01% | 0.00% | 0.00% | 0.00% | 0.00% | 0.00% |
| **[0.20, 0.40)** | 0.00% | 0.00% | 0.06% | 0.00% | 0.00% | 0.00% | 0.00% | 0.00% |
| **[0.40, 0.60)** | 0.00% | 0.00% | 0.05% | 0.00% | 0.00% | 0.00% | 0.00% | 0.00% |
| **[0.60, 0.80)** | 0.00% | 0.00% | 0.00% | 0.00% | 0.00% | 0.00% | 0.00% | 0.00% |
| **[0.80, 1.00)** | 0.00% | 0.00% | 0.00% | 0.00% | 0.00% | 0.00% | 0.00% | 0.00% |

*Table 15.* Performance of LLAMA3-8B-INSTRUCT by ablating pre-selection and NTK selection.

|  | MMLU-Med | FPB | ContractNLI | RSDD |
|---|---|---|---|---|
| NTK-Selector ($M = 4N$, default) | 73.7 | 85.5 | 70.6 | 96.5 |
| Only Pre-selection ($M = N$) | 73.0 | 85.1 | 64.6 | 94.3 |
| Pre-selection + NTK selection ($M = 2N$) | 73.3 | 84.5 | 68.9 | 95.5 |
| Pre-selection + NTK selection ($M = 16N$) | 74.4 | 86.9 | 70.6 | 95.9 |
| Only NTK selection ($M = |\mathcal{C}|$) | 74.6 | 87.0 | 71.7 | 96.7 |

*Table 16.* Performance of QWEN3-8B on agricultural irrigation QA task (AgXQA), solar cell QA task (SCQA), open-ended climate QA task (ClimaQA Freeform), and Kinyarwanda language news classification task (KINNEWS).

|  | AgXQA | SCQA | ClimaQA Freeform | KINNEWS |
|---|---|---|---|---|
| Base | 69.7 | 46.8 | 77.9 / 27.0 | 54.5 |
| Domain-only | 69.8 | 52.3 | 79.9 / 35.1 | 51.8 |
| Random | 72.1 | 64.0 | 78.4 / 27.6 | 56.9 |
| DSIR | 68.7 | 63.0 | 73.5 / 18.5 | 65.6 |
| LESS | 73.4 | 64.1 | 73.5 / 18.5 | 65.6 |
| TSDS | 76.0 | 63.5 | 77.7 / 29.1 | 67.6 |
| **NTK-Selector** | **78.3** | **64.5** | **81.3 / 40.9** | **70.8** |

uninformative answers. This makes it difficult to align auxiliary samples with the model's actual reasoning or analysis patterns. Adding CoT rationales mitigates this issue by giving a more faithful representation of how the model is expected to "think" on the task.

Importantly, within each task listed before, all methods (Domain-only, baselines, NTK-Selector) use the same set of generated CoT-annotated domain instances. The synthetic CoT is therefore part of the shared supervision signal, not a special advantage for NTK-Selector. In fact, we often see that CoT annotation also improves the Domain-only baseline, since richer rationales help the model learn better reasoning even without auxiliary data. As a result, the relative performance gains of NTK-Selector over other methods are not driven by CoT generation itself, but by how effectively each method chooses auxiliary data given the same domain-CoT supervision.

### H.10. Ablation over Selection Pipeline

In this part, we conduct an ablation over the selection pipeline on different domains where the selection size $N$ is set to be 9000 by default. As shown in Table 15, adding NTK selection on top of pre-selection brings clear additional gains over using pre-selection alone (e.g., from 64.6 to 71.7 on ContractNLI), and these gains generally increase as $M$ grows. We believe these results suggest that the NTK component provides a complementary signal beyond embedding-based semantic similarity.

### H.11. Results on Other Low-resource Tasks

To further evaluate the efficacy of NTK-Selector in domains that are less commonly studied, we introduce a low-resource language task: Kinyarwanda language new classification (KINNEWS (Niyongabo et al., 2020)). Besides, to introduce more diversified QA formats, we add extraction QA tasks ranging from agricultural irrigation (AgXQA (Kpodo et al., 2024)) to solar cell QA (SCQA (Li & Cole, 2025)). Additionally, we also introduce an open-ended QA task about climate QA (ClimaQA Freeform (Manivannan et al., 2025)). We report F1 scores for all AgXQA, SCQA, and KINNEWS, and report BertScore and Rouge-L for ClimaQA Freeform. As shown in Table 16, NTK-Selector continues to perform strongly and typically outperforms all baselines, suggesting that our approach remains effective in genuinely low-resource, specialized domains across extractive and free-form QA as well as classification tasks.

---

**FPB & TFNS**

---

【Input】
  {news_text}: Teollisuuden Voima Oyj , the Finnish utility known as TVO , said it shortlisted Mitsubishi Heavy s EU-APWR model along with reactors from Areva , Toshiba Corp. , GE Hitachi Nuclear Energy and Korea Hydro & Nuclear Power Co.
  {question_text}: What is the sentiment of this news?\nA.negative\nB.neutral\nC.positive
  {answer_text}: B
【Instruction】
  {instruction_text}: {news_text}\n{question_text}\n
【System Prompt】
  You are a financial analyst with expertise in sentiment analysis.
【User Prompt】
  Analyze the sentiment of this financial news article by following these steps:
  1. Identify key financial terms and market indicators
  2. Examine contextual clues and narrative tone
  3. Assess potential market implications
  4. Determine overall sentiment

  Begin with "Let's analyze this financial news step by step:" and conclude with "Therefore, the sentiment is [original_answer]".
  Keep the analysis professional and finance-focused.

  Instruction: {instruction_text}
  Answer: {answer_text}
【Generated Answer】
  Let's analyze this financial news step by step:\n\n1. **Identify key financial terms and market indicators**: The article mentions Teollisuuden Voima Oyj (TVO), a Finnish utility, and highlights its shortlist of reactor models from notable companies such as Mitsubishi Heavy, Areva, Toshiba Corp., GE Hitachi Nuclear Energy, and Korea Hydro & Nuclear Power Co. Key terms include \"shortlisted,\" \"reactors,\" and the names of established companies in the nuclear energy sector.\n\n2. **Examine contextual clues and narrative tone**: The tone of the article is factual and straightforward. It reports on the actions of TVO without expressing any overtly positive or negative opinions. The use of terms like \"shortlisted\" suggests a selection process but does not indicate a definitive outcome or preference, which contributes to a neutral tone.\n\n3. **Assess potential market implications**: The news indicates that TVO is in the process of evaluating options for nuclear reactors, which could have implications for the companies mentioned. However, without further information on the decision-making process or the criteria for selection, it is difficult to ascertain any immediate market impact. The neutrality of the announcement suggests that it may not significantly affect stock prices or investor sentiment at this stage.\n\n4. **Determine overall sentiment**: Given the factual nature of the report, the absence of strong language indicating optimism or pessimism, and the focus on the shortlist of options rather than a decisive action, the overall sentiment can be classified as neutral.\n\nTherefore, the sentiment is B.neutral.\nTherefore, the answer is B.

---

*Figure 9.* Query prompt and example of generated answer for financial sentiment analysis (FPB and TFNS) tasks.

## I. Query Prompts

For the tasks as detailed in Appendix G.1, the corresponding instruction datasets often contain very brief responses that lack a reasoning process, such as single-character choices (e.g., "A", "B", "C") or several words. This brevity limits a model's ability to learn complex reasoning chains and generate detailed explanations crucial for improving task accuracy. To guide the generation process, we designed a specific prompt template that incorporates the original instruction dataset. We then used this template to query GPT-4o-mini (Hurst et al., 2024), as demonstrated in Figures 9, 10, 11, 12, and 13. Each template comprises three main components: a persona-defining system prompt (e.g., "You are a financial analyst..."), a multi-step instructional prompt outlining the required analysis (e.g., "Analyze the sentiment..."), and placeholders for the original instruction and right answer. By leveraging these structured prompts, we consistently generated outputs that adhered to a predetermined format and analytical style.

---

**Headline**

【Input】
    {headline_text}: london shakes up gold hoarders' list
    {answer_text}: B
    {question_text}:Does the news headline talk about price going down?\nA.Yes\nB.No
【Instruction】
    {instruction_text}: {headline_text}\n{question_text}\n
【System Prompt】
    You are a financial analyst with expertise in news headline classification.
【User Prompt】
    Analyze whether this financial news headline discusses a price decline by following these steps:
    1. Identify key terms related to price movement or market changes
    2. Examine the context and implied action in the headline
    3. Consider the entities involved and their potential reactions
    4. Determine if there is a direct or implied mention of prices going down

    Begin with "Let's analyze this financial news step by step:" and conclude with "Therefore, the answer is [original_answer]."
    Keep the analysis professional and finance-focused.

    Instruction: {instruction_text}
    Answer: {answer_text}
【Generated Answer】
    Let's analyze this financial news step by step:\n\n1. **Identify key terms related to price movement or market changes**: The headline \"london shakes up gold hoarders' list\" does not contain any explicit terms related to price movement, such as \"decline,\" \"drop,\" \"fall,\" or \"decrease.\" Instead, it suggests a change in the ranking or status of gold hoarders, which does not inherently imply a price movement.\n\n2. **Examine the context and implied action in the headline**: The phrase \"shakes up\" indicates a disruption or change in the status quo regarding gold hoarders. However, it does not provide any specific information about the prices of gold or any related commodities. The context seems to focus more on the organizational or ranking aspect rather than market pricing.\n\n3. **Consider the entities involved and their potential reactions**: The headline mentions \"gold hoarders,\" which could refer to individuals or institutions that hold significant amounts of gold. While changes in their rankings may affect market perceptions or strategies, the headline does not suggest that these changes are linked to a decline in gold prices or any financial losses.\n\n4. **Determine if there is a direct or implied mention of prices going down**: There is no direct or implied mention of prices going down in the headline. The focus is on the list of gold hoarders rather than on the price of gold itself. Without any indication of price movements, we cannot conclude that the headline discusses a price decline.\n\nTherefore, the answer is B.

*Figure 10.* Query prompt and example of generated answer for financial headline classification (Headline) task.

---

**ContractNLI**

【Input】
{premise_text}: The aforesaid obligations of the receiving party shall survive the termination of this Agreement. 8. LIMITED WARRANTY Each party hereto warrants that it has the right to disclose the Proprietary/Confidential Information which it discloses to the other parties and that the Proprietary/Confidential Information disclosed is to the best of its knowledge, correct. Nothing contained in this agreement shall be construed to obligate any party to disclose any information to the other parties. 9. REMEDY FOR BREACH It is understood and agreed between the parties that any breach of the obligations of confidentiality contained in this Agreement may cause the disclosing party irreparable loss. Accordingly, and in addition to any other remedies a party may have in law or equity, the disclosing party shall be entitled to obtain injunctive relief against the receiving party to prevent any further or continuing breach of the receiving party's obligations or additional damage to the disclosing party in the event such loss is in fact incurred by the disclosing party as a result of the breach or is imminent. 10. SEVERABILITY If, for any reason, a court of competent jurisdiction finds any provision of this Agreement, or any portion thereof, to be unenforceable, such decision shall not affect the validity of the remaining portion, which remaining portion shall continue in full force and effect as if this Agreement had been executed with the invalid portion thereof eliminated therefrom. In the event that a portion of this Agreement shall be declared to be invalid, then the parties agree, that they shall, in good faith, negotiate with one another to replace such invalid provision with a valid provision as similar as possible to that which had been said to be invalid. 11. TERMINATION This Agreement shall govern all matters referred to herein until terminated by either party upon thirty days written notice to the other or in accordance with this Agreement. Upon termination, all information and materials shall be returned to the respective parties. Notwithstanding the termination, each party shall continue to fulfill its obligations hereunder for a period of five (5) years thereafter.
{hypothesis_text}: Some obligations of Agreement may survive termination of Agreement.
{question_text}: Given the following clause from a Non-Disclosure Agreement (premise), determine whether the hypothesis is entailed by the premise, contradicts it, or is unrelated.\nA. entailment\nB. contradiction\nC. neutral\n
{answer_text}: A
【Instruction】
{question_text}\nPremise:\n{premise_text}\nHypothesis:\n{hypothesis_text}
【System Prompt】
You are a legal reasoning assistant with expertise in regulatory analysis and corporate law.
【User Prompt】
Analyze the logical relationship between the contract clause and the hypothesis by following these steps:
1. Identify key obligations, permissions, and conditions in the clause
2. Examine the scope and intent of the hypothesis
3. Compare the language and meaning of both texts
4. Determine whether the hypothesis is entailment, contradiction, or neutral

Begin with "Let's analyze this legal text step by step:" and conclude with "Therefore, the answer is". Keep the analysis professional and law-focused.

Please make sure the result generated by the chain-of-thought leads to the correct answer, as if you are seeing this case for the first time.

Instruction: {instruction_text}
Answer: {answer_text}
【Generated Answer】
Let's analyze this legal text step by step:\n\n1. **Identify key obligations, permissions, and conditions in the clause**:\n - The obligations of the receiving party regarding confidentiality survive the termination of the Agreement.\n - Each party warrants that it has the right to disclose the Proprietary/Confidential Information and that such information is correct to the best of its knowledge.\n - No party is obligated to disclose any information to the other parties.\n - Breach of confidentiality may cause irreparable loss to the disclosing party, allowing for injunctive relief.\n - If any provision is found unenforceable, the remaining provisions remain valid, and the parties will negotiate a replacement.\n - The Agreement can be terminated by either party with thirty days written notice, and upon termination, all information must be returned. Obligations continue for five years post-termination.\n\n2. **Examine the scope and intent of the hypothesis**:\n - The hypothesis likely posits a scenario regarding the enforceability of the confidentiality obligations or the consequences of a breach of such obligations.\n\n3. **Compare the language and meaning of both texts**:\n - The clause explicitly states that confidentiality obligations survive termination, indicating a strong intent to protect proprietary information even after the Agreement ends.\n - The warranty of rights to disclose information and the acknowledgment of potential irreparable loss suggest that the parties recognize the importance of confidentiality and the potential consequences of breaches.\n\n4. **Determine whether the hypothesis is entailment, contradiction, or neutral**:\n - If the hypothesis suggests that the confidentiality obligations are enforceable and that breaches can lead to significant consequences (like injunctive relief), it aligns with the language of the clause, indicating that it is an entailment.\n - If the hypothesis suggests that confidentiality obligations do not survive termination or that breaches do not have serious consequences, it would contradict the clause.\n\nGiven the analysis, if the hypothesis supports the enforceability of confidentiality obligations and acknowledges the potential for irreparable harm due to breaches, it is consistent with the clause. \n\nTherefore, the answer is A.\n

33

*Figure 11.* Query prompt and example of generated answer for legal natural language inference (ContractNLI) task.

---

**PrivacyQA**

【Input】
  {clause_text}: Other times, we combine Usage Data collected from Cookies with that of other customers to improve your and other customers' experience.
  {question_text}: have you had any data breaches in the last couple of years?
  {answer_text}: A
【Instruction】
  {instruction_text}: Classify if the clause is relevant to answering the question.\nA. Irrelevant\nB. Relevant\n\nClause: {clause_text}\n{question_text}\n
【System Prompt】
  You are a legal expert with expertise in privacy policy analysis and question answering.
【User Prompt】
  Analyze the policy text by following these steps:
  1. Identify key obligations, permissions, and conditions mentioned in the policy segment. What does the policy explicitly state about user data? Ensure your analysis aligns with the correct answer provided at the end.
  2. Examine the scope and intent of the user question. What specific information is the user seeking? Cross-check your interpretation against the correct answer to ensure consistency.
  3. Compare the language and meaning of both the policy segment and the question. Are there any direct or indirect references in the policy that address the question's intent? Verify that your comparison matches the reasoning in the correct answer.
  4. Based on your comparison, determine if the policy segment provides enough information to answer the question. Is it directly related, indirectly related, or unrelated? Finalize your conclusion **only after confirming it matches the correct answer**.

  Begin with "Let's analyze this legal text step by step:" and conclude with "Therefore, the answer is [original_answer]". Keep the analysis professional and law-focused.

  Instruction: {instruction_text}
  Answer: {answer_text}
【Generated Answer】
  Let's analyze this legal text step by step:\n\n1. **Identify key obligations, permissions, and conditions mentioned in the policy segment.** \n The policy segment states that the company combines Usage Data collected from Cookies with that of other customers to enhance the experience for both the user and other customers. This indicates a permission to use data for improvement purposes but does not specify any obligations regarding data security or breaches.\n\n2. **Examine the scope and intent of the user question.** \n The user is specifically inquiring about data breaches that may have occurred in the last couple of years. This question seeks information about the company's data security practices and any incidents that may have compromised user data.\n\n3. **Compare the language and meaning of both the policy segment and the question.** \n The policy segment discusses the use of Usage Data and Cookies for improving user experience but does not mention anything about data breaches, security incidents, or the handling of data in the context of breaches. There are no direct or indirect references in the policy that address the user's question about data breaches.\n\n4. **Determine if the policy segment provides enough information to answer the question.** \n The policy segment is unrelated to the question about data breaches. It does not provide any information regarding the occurrence of breaches or the company's security practices. Therefore, it does not help in answering the user's inquiry.\n\nTherefore, the answer is A. Irrelevant.\n

*Figure 12.* Query prompt and example of generated answer for privacy policy-based question-answering (PrivacyQA) task.

---

**RSDD**

【Input】
  {post_text}: hi r anxiety i m a 0yr old male with what i would say is a decent amount of stress but nothing out of the norm yeah work is stressful rent is increasing and a proposal to my gf are all looming but i ve always been able to accept those thing will come with some mild stress since i m human i ve been experiencing what i can only describe a mild severe panic attack recently that are completely wiping me off the face of the earth this past saturday i found myself in a bathroom stall at noon after being out with friend for lunch drink experiencing symptom last night after work i came home wa slapped with tunnel vision trembling tight chest and once i laid down in bed had what i would describe a the worst experience of my life for 0 minute borderline exorcism i m completely new to this experience and am obviously concerned what my trigger are or where this is coming from i plan on having a wellness check with my local physician but are there buzzword or thing i should include so i can do exactly explain what i am experiencing i constantly drink water i take only about 00mg caffeine a day i exercise time a week i have what i would describe a a pretty well balanced diet why is this happening all of a sudden appreciate any feedback this is more of a vent
  {question_text}: Classify whether a user has depression based on their Reddit posts and choose an option:\nA. Yes\nB. No
  {answer_text}: A
【Instruction】
  {instruction_text}: {question_text}\nUser post: {post_text}
【System Prompt】
  You are a clinical psychologist with expertise in mood disorders and digital behavioral analysis.
【User Prompt】
  Analyze the post by following these steps:
  1. Identify key phrases indicating depressive symptoms or mental state
  2. Examine behavioral patterns described (sleep habits, avoidance, etc.)
  3. Evaluate expressed emotions and cognitive patterns
  4. Consider diagnostic criteria for depression (DSM-5/ICD-10)
  5. Determine if evidence supports depression classification

  Begin with "Let's analyze this Reddit post step by step:" and conclude with "Therefore, the answer is [original_answer]". Keep the analysis clinically focused and evidence-based.
  Please make sure the result generated by the chain-of-thought leads to the given correct answer.

  Instruction: {instruction_text}
  Answer: {answer_text}
【Generated Answer】
  Let's analyze this Reddit post step by step:\n\n1. **Identify key phrases indicating depressive symptoms or mental state**: The user describes experiencing \"mild severe panic attacks,\" \"tunnel vision,\" \"trembling,\" and a \"tight chest.\" Phrases like \"completely wiping me off the face of the earth\" and \"the worst experience of my life\" suggest a significant level of distress. The mention of being \"obviously concerned\" about the triggers indicates a heightened level of anxiety and potential hopelessness.\n\n2. **Examine behavioral patterns described (sleep habits, avoidance, etc.)**: The user does not explicitly mention sleep habits, but the context implies that the panic attacks are affecting their overall functioning. The user describes a situation where they had to retreat to a bathroom stall, suggesting avoidance behavior in response to overwhelming anxiety. This indicates a disruption in daily activities and social interactions.\n\n3. **Evaluate expressed emotions and cognitive patterns**: The user expresses feelings of fear and confusion about their sudden panic attacks. They mention feeling \"completely new to this experience,\" which indicates a lack of coping strategies for managing their anxiety. The cognitive pattern appears to be focused on worry about the future (work stress, rent increase, proposal), which can exacerbate feelings of anxiety and helplessness.\n\n4. **Consider diagnostic criteria for depression (DSM-5/ICD-10)**: According to the DSM-5, symptoms of depression can include persistent sadness, loss of interest or pleasure, feelings of worthlessness, and difficulty concentrating. While the user primarily describes anxiety symptoms, the distress and functional impairment they report could indicate a comorbid depressive episode, especially given the overwhelming nature of their panic attacks.\n\n5. **Determine if evidence supports depression classification**: The user's description of their experiences suggests significant distress and impairment in functioning, which aligns with depressive symptoms. The combination of anxiety and the emotional turmoil expressed could support a diagnosis of depression, particularly if these symptoms persist over time.\n\nTherefore, the answer is A.\n

*Figure 13.* Query prompt and example of generated answer for depression diagnosis (RSDD) task.

## J. Case Study of Selected Auxiliary Samples

In this part, we demonstrate examples of the domain sample and the selected auxiliary sample by NTK-Selector for each task as shown in Figures 14, 15, and 16, showcasing their shared topics, reasoning patterns, or inherent capabilities. For instance, in the ContractNLI task, although the domain sample's topic (non-disclosure agreements) and the auxiliary sample's topic (constitutional issues) differ, both are textual entailment tasks and require a similar reasoning pattern to determine the relationship between the premise and the hypothesis. This example highlights that our selector effectively identifies samples that share underlying logical structures, even when their semantic content varies.

---

**MedMCQA & MMLU-Med**

【Domain Sample】

{instruction}: In patients with pancreas divisum, symptomatic obstructive pancreatopathy, and contraindications for immediate definitive surgery, which intervention is considered a bridge therapy to alleviate symptoms while awaiting a future surgical or endoscopic resolution?\nA. Lesser papilla sphincteroplasty\nB. Endoscopic stenting through the lesser papilla\nC. Decompressive operation\nD. Total pancreatectomy\n

{answer}: Pancreas divisum is a common congenital anomaly where the ductal systems of the dorsal and ventral pancreas fail to fuse during embryonic development. This can lead to a scenario where the majority of the pancreatic secretions must flow through the smaller duct of Santorini and the minor papilla, rather than through the duct of Wirsung and the major papilla. In some patients, this can cause recurrent episodes of pancreatitis due to the difficulty in draining pancreatic secretions.\n\nWhen patients with pancreas divisum present with symptomatic obstructive pancreatopathy and are not immediate candidates for definitive surgery, the goal is to relieve the obstruction and alleviate symptoms until a more permanent solution can be achieved.\n\nA. Lesser papilla sphincteroplasty is a surgical procedure that involves cutting the sphincter muscle to allow better drainage of the pancreatic duct. However, this is a form of definitive surgery and may not be indicated if there are contraindications for immediate surgery.\n\nB. Endoscopic stenting through the lesser papilla involves placing a stent endoscopically to help drain the pancreatic duct. This is less invasive than surgery and can be used as a temporary measure to relieve symptoms.\n\nC. Decompressive operation is a surgical intervention aimed at relieving pressure or obstruction, but again, this is a form of definitive surgery and may not be suitable if surgery is contraindicated at the moment.\n\nD. Total pancreatectomy is the complete surgical removal of the pancreas and is a definitive and radical treatment for chronic pancreatitis or other severe pancreatic diseases. It is not a bridge therapy and is associated with significant morbidity, including the development of diabetes mellitus.\n\nGiven the need for a bridge therapy that is less invasive and can alleviate symptoms while awaiting future definitive resolution, endoscopic stenting through the lesser papilla (B) is the most appropriate intervention among the options listed. It can provide symptomatic relief and is reversible, serving as a bridge until the patient is fit for surgery or another definitive endoscopic treatment.\n\nSo, the answer is B.

【Selected Auxiliary Sample】

{instruction}: What is the most common symptom of MI?\n\nIs The most common symptom is chest pain or discomfort which may travel into the shoulder, arm, back, neck, or jaw. a good answer to this question?\n\nOPTIONS:\n- yes\n- no\n

{answer}: The statement provided (\"the most common symptom is chest pain or discomfort which may travel into the shoulder, arm, back, neck, or jaw\") accurately answers the question \"What is the most common symptom of MI?\" So, in this case the answer would be \"yes\".

---

**FPB & TFNS**

【Domain Sample】

{instruction}: Starbucks says the workers violated safety policies while workers said they'd never heard of the policy before and are alleging retaliation.\nWhat is the sentiment of this news?\nA.strong negative\nB.moderately negative\nC.mildly negative\nD.neutral\nE.mildly positive\nF.moderately positive\nG.strong positive\n\n

{answer}: Let's analyze this financial news step by step:\n\n1. **Identify key financial terms and market indicators**: The article mentions \"Starbucks,\" which is a publicly traded company, and refers to \"workers\" and \"safety policies.\" The mention of safety policies and allegations of retaliation can indicate potential labor disputes, which may affect the company's operational efficiency and public perception.\n\n2. **Examine contextual clues and narrative tone**: The tone of the article appears to be contentious, highlighting a conflict between the company and its workers. The phrase \"violated safety policies\" suggests a serious issue, while the workers' claim that they were unaware of these policies introduces a sense of confusion and potential miscommunication. The mention of \"retaliation\" adds a layer of severity to the situation, indicating possible legal implications.\n\n3. **Assess potential market implications**: Labor disputes and allegations of retaliation can lead to negative publicity for Starbucks, potentially affecting its stock price and investor sentiment. If the situation escalates, it could result in operational disruptions or increased costs related to legal proceedings and public relations efforts. Such factors may deter investors and impact the company's market performance.\n\n4. **Determine overall sentiment**: Considering the contentious nature of the article, the serious implications of the allegations, and the potential negative impact on Starbucks' reputation and operations, the overall sentiment can be characterized as moderately negative.\n\nTherefore, the sentiment is B. moderately negative.\nTherefore, the answer is B.

【Selected Auxiliary Sample】

{instruction}: Given a piece of financial news and its polarity, classify it into 'true' if the polarity is correct and classify into 'false' if the polarity is incorrect. Output must be 'true' or 'false'. \n\nnews:The acquisition price was not disclosed .\npolarity:neutral\n

{answer}: The polarity of the news is neutral. This means that the news does not convey any positive or negative sentiment towards the subject. In this example, the financial news is about an acquisition for which no price has been disclosed, and hence it does not convey any sentiment towards that acquisition. So, this financial news can be classified as \"true\".

*Figure 14.* Example of domain sample and selected auxiliary sample of medical question answering (MedMCQA and MMLU-Med) and financial sentiment analysis (FPB and TFNS) tasks.

---

**Headline**

【Domain Sample】
{instruction}: Gold prices settle lower after two-session gain\nDoes the news headline talk about price staying constant? \nA.Yes\nB.No\n

{answer}: Let's analyze this financial news step by step:\n\n1. **Identify key terms related to price movement or market changes**: The headline contains the term \"settle lower,\" which directly indicates a decrease in price. Additionally, \"two-session gain\" suggests that there was a previous increase in price before the current decline.\n\n2. **Examine the context and implied action in the headline**: The phrase \"settle lower\" implies that the price of gold has decreased after experiencing gains in the previous sessions. This indicates a reversal in price movement, suggesting that the market has reacted negatively after a period of increase.\n\n3. **Consider the entities involved and their potential reactions**: The entities involved here are likely investors and traders in the gold market. A decline in gold prices may lead to reactions such as selling off positions, reevaluating investment strategies, or increased market volatility as traders respond to the new price level.\n\n4. **Determine if there is a direct or implied mention of prices going down**: The headline explicitly states that gold prices have \"settled lower,\" which is a direct indication of a price decline. There is no ambiguity in this phrasing; it clearly communicates that prices have decreased.\n\nTherefore, the answer is B.

【Selected Auxiliary Sample】
{instruction}: Classify the given a piece of financial news into three classes: positive, negative, and neutral. Output must be 'positive', 'negative', or 'neutral'.\n\nSubscription sales decreased slightly .\n

{answer}: The financial news states that subscription sales have decreased. This indicates a drop in sales, which is generally considered as a negative development. So, this financial news can be classified as \"negative\".\n\"\"

---

**ContractNLI**

【Domain Sample】
{instruction}: Given the following clause from a Non-Disclosure Agreement (premise), determine whether the hypothesis is entailed by the premise, contradicts it, or is unrelated.\nA. entailment\nB. contradiction\nC. neutral\n\nPremise:\nEach party agrees that all Confidential Information of the other party, including any copies thereof, shall be returned to the disclosing party upon request or destroyed within ten (10) business days of the expiration or other termination of the Project. However, notwithstanding anything to the contrary herein, 1) neither party will be required to delete electronic Confidential Information stored in back-up/archival storage in accordance with its policies, provided that any such retained Confidential Information will continue to be subject to the terms of this Confidentiality Agreement until it is destroyed by the receiving party; and 2) a single copy of all Confidential Information may be retained by the Receiving Party\u2019s legal department for dispute resolution purposes only, provided, however, that such Confidential Information shall remain subject to the terms and conditions of this Confidentiality Agreement. In the event that a receiving party has or acquires actual knowledge of any breach of the confidentiality of, or the misappropriation of, any Confidential Information received under this Confidentiality Agreement, such party shall promptly give notice thereof to the other party. 5. Upon written demand by the disclosing party or upon termination of this Confidentiality Agreement, the receiving party shall: (i) cease using the Confidential Information, (ii) return the Confidential Information and all copies, notes or extracts thereof to the disclosing party within ten (10) business days of receipt of demand; and \nHypothesis:\nAll Confidential Information shall be expressly identified by the Disclosing Party.\n

{answer}: Let's analyze this legal text step by step:\n\n1. **Identify key obligations, permissions, and conditions in the clause**:\n - Both parties are required to return or destroy all Confidential Information upon request or within ten business days after the termination of the Project.\n - There are exceptions to this obligation:\n - Electronic Confidential Information stored in back-up/archival storage does not need to be deleted immediately, but it must remain subject to the confidentiality terms until destroyed.\n - A single copy of Confidential Information may be retained by the Receiving Party\u2019s legal department solely for dispute resolution purposes, with the condition that it remains subject to the confidentiality terms.\n - If either party becomes aware of a breach of confidentiality, they must promptly notify the other party.\n - Upon written demand or termination of the Confidentiality Agreement, the receiving party must cease using the Confidential Information and return it within ten business days.\n\n2. **Examine the scope and intent of the hypothesis**:\n - The hypothesis likely posits a scenario regarding the obligations of the parties concerning the handling of Confidential Information, particularly in the context of breaches or the return of information.\n\n3. **Compare the language and meaning of both texts**:\n - The clause outlines specific obligations regarding the return and destruction of Confidential Information, as well as the conditions under which certain information may be retained. The language is clear about the responsibilities of both parties and the exceptions that apply.\n - If the hypothesis suggests that one party can disregard these obligations or that the confidentiality terms do not apply to retained information, it would contradict the explicit terms of the clause.\n\n4. **Determine whether the hypothesis is entailment, contradiction, or neutral**:\n - If the hypothesis aligns with the obligations and exceptions outlined in the clause, it would be considered entailment. If it suggests that the obligations can be ignored or that the confidentiality terms do not apply, it would be a contradiction. If it presents a scenario that does not clearly align with the obligations or exceptions, it would be neutral.\n\nGiven the analysis, if the hypothesis contradicts the obligations and conditions set forth in the clause regarding the handling of Confidential Information, then it is a contradiction.\n\nTherefore, the answer is C.\n

【Selected Auxiliary Sample】
{instruction}: Premise: The court found it unconstitutional.\nHypothesis: It was found to be unconstitutional by the court.\n\nDoes the premise entail the hypothesis?\n\nOPTIONS:\n- yes\n- it is not possible to tell\n- no\n

{answer}: The premise specifically states that the court found it unconstitutional. This implies that what \"it\" refers to was declared unconstitutional by the court, which is exactly what is stated in the hypothesis. Hence, we can conclude that this statement entails the hypothesis and so choose option A: yes

---

**PrivacyQA**

【Domain Sample】

{instruction}: Classify if the clause is relevant to answering the question.\nA. Irrelevant\nB. Relevant\n\nClause: For personal data subject to the European Union General Data Processing Regulations (GDPR), we rely on several legal bases to process the data.\nQuestion: can any one access my private information collected by the app?\n

{answer}: Let's analyze this legal text step by step:\n\n1. **Identify key obligations, permissions, and conditions mentioned in the policy segment.** The policy states that for personal data governed by the GDPR, the organization relies on several legal bases to process that data. This implies that there are specific legal frameworks and conditions under which user data can be processed, but it does not explicitly address who can access the data or under what circumstances.\n\n2. **Examine the scope and intent of the user question.** The user is seeking information about whether anyone can access their private information collected by the app. This question is focused on the accessibility of personal data and who has the right to access it, which is not directly addressed in the provided policy segment.\n\n3. **Compare the language and meaning of both the policy segment and the question.** The policy segment discusses the legal bases for processing personal data under GDPR but does not mention access rights or who may access the data. The user question specifically inquires about access to private information, which is not covered in the policy text. Therefore, there is no direct or indirect reference in the policy that addresses the user's concern about access to their private information.\n\n4. **Determine if the policy segment provides enough information to answer the question.** Given that the policy does not mention anything about access to private information or who can access it, it is clear that the clause is unrelated to the user's question. \n\nTherefore, the answer is A. Irrelevant.\n

【Selected Auxiliary Sample】

{instruction}: In this task, you are given a statement spoken by a politician in natural language and the subject of the discussion for the given statement. Your task is to determine if the given subject of the discussion is correct for the given statement. The subject can be multiple words. Answer with \"True\" if it's the correct subject of discussion or return \"False\" if it's incorrect.\n\nStatement: On banning assault weapons. \n Subject: guns\n

{answer}: The subject of discussion in the statement is guns and we can see that the politician is talking about banning assault weapons. Therefore, the answer to this question should be True.

---

**RSDD**

【Domain Sample】

{instruction}: Classify whether a user has depression based on their Reddit posts and choose an option:\nA. Yes\nB. No\nUser post: i wa scrolling around and read up on brain aneurysm and im terrified of having one now i saw a headache wa a sign and i read that while having one it not severe but still now im on the brink of cry and i wan na ask my parent for a check up because of this anxiety but i know they aren t willing to do it can someone offer some relief\n

{answer}: Let's analyze this Reddit post step by step:\n\n1. **Identify key phrases indicating depressive symptoms or mental state**: The user expresses feelings of terror and anxiety about health concerns, stating they are \"on the brink of cry.\" This suggests a heightened emotional state that may be indicative of distress. The phrase \"I know they aren\u2019t willing to do it\" indicates a sense of helplessness or lack of support, which can contribute to depressive feelings.\n\n2. **Examine behavioral patterns described (sleep habits, avoidance, etc.)**: While the user does not explicitly mention sleep habits, their anxiety about health issues may lead to avoidance behaviors, such as not wanting to discuss their fears with their parents. The act of scrolling through potentially distressing information about brain aneurysms suggests a compulsive behavior that can be linked to anxiety, which often coexists with depressive symptoms.\n\n3. **Evaluate expressed emotions and cognitive patterns**: The user expresses significant anxiety and fear regarding their health, which can lead to rumination\u2014a cognitive pattern often associated with depression. The desire to cry indicates a level of emotional distress that can be symptomatic of depression. The user's plea for relief suggests feelings of desperation and a need for support.\n\n4. **Consider diagnostic criteria for depression (DSM-5/ICD-10)**: According to the DSM-5, symptoms of depression can include persistent sadness, feelings of hopelessness, and anxiety. The user's emotional state, coupled with their avoidance of seeking help from their parents, aligns with the criteria for mood disorders. The distress caused by their health anxiety could also contribute to a depressive episode.\n\n5. **Determine if evidence supports depression classification**: The combination of expressed anxiety, emotional distress, and avoidance behavior suggests that the user is experiencing significant psychological strain. While they may not meet all criteria for a major depressive episode, the symptoms they describe indicate a likelihood of experiencing depressive feelings, particularly in the context of their anxiety.\n\nTherefore, the answer is A.\n

【Selected Auxiliary Sample】

{instruction}: Categorize the comment on the basis of threat. If the comment is threatening output Yes, otherwise output No.\n\nGreat\n

{answer}: The comment is a positive response and does not contain any threatening language. Therefore, the answer is \"No\".

---

*Figure 16.* Example of domain sample and selected auxiliary sample of privacy policy question-answering (PrivacyQA) and depression diagnosis (RSDD) tasks.

