# OpenReview forum: "Mining Useful General Data for Low-Resource Domain Adaptation"
_ICML.cc/2026/Conference — ICML 2026 regular_

### Official Review · Reviewer_gUWb · 2026-03-12

**Soundness:** 3
**Presentation:** 3
**Significance:** 3
**Originality:** 3
**Overall Recommendation:** 5
**Confidence:** 4

**Summary:**

This paper proposes NTK-Selector, an auxiliary data selection method for low-resource LLM fine-tuning based on neural tangent kernel (NTK) signals. The key idea is that although the NTK of a pretrained LLM is not exactly constant during fine-tuning, it remains highly aligned with its initialization, making the initial kernel a useful proxy for data utility. The paper further introduces a Jacobian-free approximation and random projection to make the method practical for modern LLMs. The empirical results are strong across several tasks and model families, and the supporting analyses on kernel stability and approximation quality are useful.

**Compliance With Llm Reviewing Policy:**

Affirmed.

**Final Justification:**

This work introduces a highly original and significant framework for low-resource domain adaptation, and the newly provided empirical evidence for $\epsilon$ stability further reinforces the validity of the core "NTK-like behavior" assumption. Ultimately, the method's ability to provide substantial performance gains through a principled, mathematically-grounded selection process makes it a valuable contribution to the field.

**Key Questions For Authors:**

1. Do the authors have direct evidence that candidate utility rankings remain stable during training, beyond NTK cosine similarity?
2. Could the authors comment on how to detect situations in which adding auxiliary data may not be beneficial?
3. How robust is the Jacobian-free approximation in more open-ended generation settings?
4. Based on the efficiency analysis, what default configuration would the authors recommend for practitioners?

I encourage the authors to address the points raised above in the weaknesses and questions, which would further improve an already strong paper.

**Limitations:**

The paper adequately discusses practical limitations such as computational overhead and the scope of the empirical evaluation. However, a slightly more explicit discussion of potential failure modes when the auxiliary candidate pool is weakly aligned with the target domain could further improve the limitations section, but overall I believe the discussion is adequate.

**Strengths And Weaknesses:**

Strengths:
The paper addresses an important and practical problem, and does so from a technically interesting perspective. I found the empirical results convincing, especially in very low-resource settings where the gains over strong baselines are meaningful. I also appreciated that the paper validates the proposed approximations rather than simply introducing them as engineering heuristics. The Jacobian-free NTK and projected version both appear to preserve the relevant ranking information very well. Overall, the paper combines novelty, empirical strength, and practical relevance well.

Weaknesses:
My main concern is about the positioning of the theory. The theoretical results provide a meaningful motivation for why the initial kernel may remain informative during fine-tuning, but I would interpret them as supportive justification rather than a full guarantee of utility ranking stability or end-to-end optimality. In addition, the success of the method seems likely to depend on the quality and transferability of the auxiliary candidate pool, and I would have liked a somewhat more explicit discussion of potential failure modes. Finally, although the proposed approximations significantly reduce cost, the method still appears more involved than simpler retrieval-based alternatives, so a clearer discussion of when the added complexity is worthwhile would strengthen the paper further.

---

> ### Author Rebuttal · Authors · 2026-03-31
>
> **We sincerely appreciate the reviewer’s careful reading and positive assessment.**
>
> > *W1: My main concern is about the positioning of the theory.*
> >
>
> Re: We agree that our theory is best viewed as supportive justification, not a full guarantee of ranking stability or end-to-end optimality. **This is due to the mismatch between classical NTK theory and realistic LLM fine-tuning**: pretrained LLMs do not satisfy ideal NTK assumptions exactly; exact NTK is infeasible and must be approximated (Jacobian-free, LoRA gradients, random projection); and the selector only observes the domain training set, not the hidden test distribution. Thus, the theory motivates the surrogate, while stronger claims are empirical (Table 3).
>
> > *W2: A more explicit discussion of potential failure modes.*
> >
>
> Re: We agree that success depends on both **candidate-pool quality** and **selector quality**. Our experiments fix the pool and compare selectors, isolating the latter. Still, **even an ideal selector will fail if the pool contains little transferable signal**; this also explains the diminishing returns in Figure 3. In practice, the pool should match the target not necessarily in domain knowledge, but in task type, reasoning pattern, or required skills. We will make these failure modes more explicit.
>
> > *W3: A clearer discussion of when the added complexity is worthwhile would strengthen the paper further.*
> >
>
> Re: NTK-Selector is indeed more involved than retrieval-style alternatives, so its value depends on the bottleneck. In our view, **retrieval-based methods and our approach address different problems.** Retrieval is most useful when the issue is missing external knowledge at inference time. Our setting is different: the model may already have broad background knowledge from pre-training, yet still lack the task-aligned reasoning pattern / supervision signal needed for low-resource adaptation. This is better addressed by improving fine-tuning data than by per-query retrieval alone. This is also consistent with our stronger gains on finance/law/psychology than on some medical tasks in Table 1.
>
> >
> >
> >
> > *Q1: Do the authors have direct evidence that candidate utility rankings remain stable during training, beyond NTK cosine similarity?*
> >
>
> Re: Yes. Beyond NTK cosine similarity, we directly measure selected-subset overlap during training. On Qwen3-8B (LoRA) for financial sentiment analysis, repeatedly selecting 100 from 400 candidates across checkpoints yields:
>
> |  | Epoch 0 | Epoch 2 | Epoch 4 | Epoch 6 | Epoch 8 | Epoch 10 |
> | --- | --- | --- | --- | --- | --- | --- |
> | Cosine similarity of NTK matrix | 1.0000 | 0.9986 | 0.9982 | 0.9979 | 0.9976 | 0.9975 |
> | Selected overlap | 100% | 99% | 98% | 97% | 97% | 96% |
>
> > *Q2: How to detect situations in which adding auxiliary data may not be beneficial?*
> >
>
> Re: Empirically, we find **the average NTK similarity of the selected subset serves as a useful proxy for whether adding more auxiliary data**. As shown in [1,2], as the pre-selection pool grows, if the average selected NTK score continues to increase, downstream accuracy also tends to improve (e.g., MMLU-Med); when it saturates, downstream gains diminish (e.g., ContractNLI). Thus, if enlarging the pool no longer raises the average NTK utility score, adding more auxiliary data is unlikely to help much.
>
> [1] https://anonymous.4open.science/r/ICML2026_rebuttal-3883/NTK_sim_acc/ContractNLI.png
>
> [2] https://anonymous.4open.science/r/ICML2026_rebuttal-3883/NTK_sim_acc/MMLU-Med.png
>
> >
> >
> >
> > *Q3: How robust is the Jacobian-free approximation in more open-ended generation settings?*
> >
>
> Re: **The Jacobian-free approximation is not tied to a specific output format**. It operates on output gradients, whether the output is a single token, a short answer, or a longer sequence. What matters is the gradient signal, not whether the task is classification-style or open-ended generation.
>
> > *Q4: What default configuration would the authors recommend for practitioners?*
> >
>
> Re: The adjustable configuration in this work contains: (1) the model to obtain the proxy NTK scores, (2) candidate pool, (3) auxiliary dataset size (4) pre-selection size, (5) projection dimension p.
>
> For practitioners, we recommend: (1) build a candidate pool aligned in task type/reasoning/skills; (2) use the downstream model as proxy when feasible (otherwise a smaller one); (3) gradually enlarge pre-selection size until the average NTK utility score saturates. In this process, projection size p=8192 is sufficient enough in our experiments.
>
>
> **We are genuinely excited about this direction and believe it addresses a meaningful and practically important problem. We also very much appreciate the reviewer’s thoughtful feedback, and would be happy to discuss any remaining questions or clarifications.**

---

> > ### Author Rebuttal · Reviewer_gUWb · 2026-04-05
> >
> > My original concerns regarding the paper's theoretical assumptions and practical scalability have been adequately addressed in the rebuttal through additional experiments and clarifications. While the computational overhead remains higher than simple embedding-based methods, the authors have successfully argued that this is a justifiable one-time cost for the significant performance gains achieved in low-resource settings. Overall, the paper presents a rigorous and innovative approach to data selection.

---

> > > ### Author Response · Authors · 2026-04-06
> > >
> > > We sincerely appreciate your recognition of our work and your encouraging assessment of the paper. It is especially encouraging to know that the additional experiments and clarifications in the rebuttal have adequately addressed your concerns. Thank you as well for the thoughtful comments that helped us improve the manuscript.

---

### Official Review · Reviewer_HJcp · 2026-03-13

**Soundness:** 2
**Presentation:** 3
**Significance:** 2
**Originality:** 2
**Overall Recommendation:** 3
**Confidence:** 4

**Summary:**

This work introduce a data selection approach based on Neural Tangent Kernel (NTK), to effectively use the general domain data for better adaptation in low-resourced target domains. Specifically, the authors use a Jacobian-free NTK approximation to compute the correlation between the data points from small domain-specific dataset $D$ and the large general-purpose candidate corpus $S_{pre}$ (after Coarse-grained pre-selection ), and then select the top-N samples from the general corpus to be added into the training corpus $S$. The authors use a coarse-to-fine selection procedure in practice. Experimental results on several domains show improvements compared with several baselines.

**Compliance With Llm Reviewing Policy:**

Affirmed.

**Final Justification:**

The authors have not addressed my concerns:

1. On the correctness of the training objective (1) in the Sec. of Problem Statement. The training objective (1) in the Problem Statement is a bi-level optimization problem, for which the solution is significantly different from the approach proposed by this work (single-level optimization). The author's response to this concern is that "We do NOT claim to solve Eq. (1) exactly". This does not make sense, because solving the problem of (1) needs a significantly different algorithm rather than the proposed approach, thus there exists a large gap between the theoretical foundation (problem statement of Sec. 2.1 in the paper) and the pratical implement (Sec. 4, the proposed method) in this paper.

2. The novelty is limited. There has existed a range of related works on gradient-based data selection (see [1-3]) and Jacobian-free approximation for NTK computation (cf. Mohamadi et al., 2022 https://arxiv.org/abs/2206.12543).

Overall, I tend to keep my original score.

**Key Questions For Authors:**

see weaknesses.

**Strengths And Weaknesses:**

### Strengths:
1. The motivation of using a data selection approach for fine-tuning LLMs in low-resource domains is clear and interesting.
2. The proposed NTK based data selection methods is interesting and easy to follow.
3. Extensive experimental results are presented to demonstrate the effectiveness.

### Weaknesses:
1. The proposed approach is very similar to gradient-based selection for efficient adaptation of neural networks/LLMs as proposed in the previous works [1-3]. However, the authors have not discussed or compared with these works in the paper.

2. The proposed approach uses a bi-level objective (1) for fine-tuning. However, they have not discussed how this objective is optimized in the paper,  and miss comparison with other works that use similar objectives.

3. The comparisons only consider data selection baselines but not consider low-resource domain adaptation approaches for LLMs, which are also be well-studied in the literature.

4.  The computation and memory costs have not been discussed and compared with the baselines in this paper.

[1] GRAD-MATCH: Gradient Matching based Data Subset Selection for Efficient Deep Model Training, ICML 2021.
[2] Gradient-Based Sample Selection for Black-Box Universal Domain Adaptation, AAAI 2025.
[3] ClusterUCB: Efficient Gradient-Based Data Selection for Targeted Fine-Tuning of LLMs, EMNLP 2025.

---

> ### Author Rebuttal · Authors · 2026-03-31
>
> **We thank the reviewer for the careful reading and thoughtful comments. We respond point by point below.**
>
> > *W1: The authors have not discussed or compared with these works [1-3] in the paper.*
> >
>
> Re: We would like to clarify this relation from following aspects:
>
> - **Methods [1–3] belong to the same broad loss-gradient-based selection family already represented by Gradient and LESS in Table 1**. In this sense, our intention was not to exclude this line of work, but we agree that the paper should make this connection clearer.
> - **Our method nevertheless differs from this family in the scoring signal it uses.** Prior gradient-based selection methods typically rely on loss gradients or influence-style approximations derived from them, whereas NTK-Selector uses output gradients and the induced NTK similarity to measure alignment in fine-tuning dynamics with complete theoretical evidence in Appendix C E, rather than first-order loss influence under a specific loss function.
> - **This distinction is also empirically meaningful.** In Table 1, both Gradient and LESS already underperform NTK-Selector. In addition, following the reviewer’s suggestion, we reproduced ClusterUCB [2] on Llama3-8B-Instruct, and its overall performance is also below our method:
>
> |  | MedMCQA | MMLU-Med | FPB | TFNS | Headline | ContractNLI | PrivacyQA | RSDD | Avg. |
> | --- | --- | --- | --- | --- | --- | --- | --- | --- | --- |
> | ClusterUCB | 58.3 | 73.7 | 82.4 | 70.2 | 86.0 | 66.9 | 66.6 | 94.6 | 74.8 |
> | NTK-Selector | **59.1** | **73.8** | **85.5** | **73.3** | **86.2** | **70.6** | **67.8** | **96.5** | **76.6** |
>
> We will revise the related-work discussion to make this connection and distinction clearer.
>
> > *W2: The proposed approach uses a bi-level objective (1) for fine-tuning. However, they have not discussed how this objective is optimized in the paper, and miss comparison with other works that use similar objectives.*
> >
>
> Re: **Eq. (1) is a formal statement of the subset selection objective, not the claim that we solve the bi-level program exactly.** Direct optimization would require enumerating $\left( ^{|\mathcal{C}|}_{|\mathcal{S}|} \right)$ subsets and training/evaluating each of them, which is intractable. As in other data selection works, the practical question is therefore how to design a tractable surrogate for this objective.
>
> Our answer is the NTK-based utility score in Sec. 4.3: we use NTK similarity between candidate and domain samples to predict which auxiliary data will be most useful after training (Lines 223-230), and then perform greedy top-N selection. The validity of this surrogate is supported by Table 3, where eNTK scores correlate well with actual downstream fine-tuning performance. We will make this distinction between the formal objective and the practical surrogate optimization more explicit in the revision.
>
> > *W3: The comparisons only consider data selection baselines but not consider low-resource domain adaptation approach approaches for LLMs, which are also be well-studied in the literature.*
> >
>
> Re: Low-resource domain adaptation methods address a different layer of the problem: **they study how to adapt given a fixed small domain dataset, while our paper asks how to construct a better training set** when domain data itself is too scarce. This is why we compare against data-selection baselines rather than generic adaptation methods.
>
> In this sense, **our method is complementary to standard low-resource adaptation pipelines** rather than interchangeable with them. In fact, all of our experiments are already conducted in a standard LoRA-based fine-tuning setup; the gain comes from improving the training data available to adaptation, not from changing the adaptation algorithm itself. We will clarify this distinction more explicitly in the revision.
>
> > *W4: The computation and memory costs have not been discussed and compared with the baselines in this paper.*
> >
>
> Re: **We do discuss efficiency in the paper**: Table 4 reports both asymptotic complexity and wall-clock runtime, and Appendix H.6 reports the memory cost. Compared with strong baselines such as LESS, NTK-Selector is substantially more efficient because LESS requires tracking gradient changes across multiple checkpoints in order to approximate trajectory-level influence, whereas our method only needs gradients from a single checkpoint.
>
> **We sincerely thank the reviewer for the constructive feedback, which has significantly helped improve the quality of our work. We remain fully committed to addressing any further questions and look forward to any additional guidance to make this study even more impactful.**

---

> > ### Author Rebuttal · Reviewer_HJcp · 2026-04-04
> >
> > The reviewer thanks the authors for the rebuttal.
> >
> > However, I have remaining concerns as follows.
> >
> > 1. "Eq. (1) is a formal statement of the subset selection objective, not the claim that we solve the bi-level program exactly" -- It seems that there is a gap between the design objective and the practical implementation. This can not address my concerns about weakness 2.
> >
> > I have a remaining question about the novelty of this paper.
> >
> > 2. Can you have a substantial comparison (e.g., from the theoretical perspective) between the gradient-based method, e.g., [1]-[3] and the proposed approach?

---

> > > ### Author Response · Authors · 2026-04-06
> > >
> > > >
> > > >
> > > >
> > > > "Eq. (1) is a formal statement of the subset selection objective, not the claim that we solve the bi-level program exactly" -- It seems that there is a gap between the design objective and the practical implementation. This can not address my concerns about weakness 2.
> > > >
> > >
> > > Re: We thank the reviewer for the clarification request. We would like to stress that **the gap between formal objectives and practical optimization is universal in task-oriented data selection**.
> > >
> > > 1. ***Why the Gap is Unavoidable*:** Direct optimization of Eq. (1) is impossible because (i) benchmark sets are unavailable during selection and (ii) exhaustive subset search is computationally infeasible. Thus, all methods must use a tractable proxy. Eq. (1) serves to formalize the task-level goal, not to claim exact optimization.
> > > 2. ***Standard Practice*:** This paradigm is shared by all prior methods:
> > >     - DiSF (ICLR'25) [4] uses spectral structure as a proxy for downstream Harness benchmark performance.
> > >     - LESS (ICLR'24) [5] approximates influence via validation set gradients.
> > >     - Grad-MATCH [1], B2-UniDA [2], and ClusterUCB [3] all use surrogate metrics (gradient matching, sample selection metric, and UCB scores).
> > >
> > >     **In this sense, the combination of a bi-level objective plus a tractable proxy is not a weakness specific to our method, but the standard formulation in data selection.**
> > >
> > > 3. ***What Our Contribution is***: We do **NOT** claim to solve Eq. (1) exactly. Our contribution is proposing a better proxy for low-resource domain adaptation. We argue and demonstrate theoretically and empirically that NTK-based alignment captures sample interactions during fine-tuning more effectively than prior proxies. We will clarify this more clearly in the revised version.
> > >
> > > > Can you have a substantial comparison (e.g., from the theoretical perspective) between the gradient-based method, e.g., [1]-[3] and the proposed approach?
> > > >
> > >
> > > Re: We thank the reviewer for this important suggestion. The key theoretical difference is that prior works [1-3] rely on loss-gradients, while we use output-gradients (the basis of NTK):
> > >
> > > $\nabla_{\theta} \mathcal{L}(f(x;\theta), y) = \nabla_{\theta} f(x;\theta)^{\top} \nabla_{f} \mathcal{L}(f(x;\theta), y)$
> > >
> > > Here, the left-hand side is the **loss-gradient** (used by [1-3]), while the right-hand side decomposes it into the **output-gradient** (our core metric) and a **loss-dependent term**.
> > >
> > > ***Key Differences between Loss-gradient and Output-gradient***:
> > >
> > > 1. **Utility Notion**: Loss-gradient $\nabla_{\theta}\mathcal{L}$ specifies a local update to minimize error on a specific label $y$. In contrast, output-gradient $\nabla_{\theta} f$ characterizes the functional sensitivity, i.e., how the model's prediction for $x$ shifts in response to parameter perturbations, serving as a "feature signature" of $x$ in the function space, independent of the current loss or label
> > > 2. **Training Dynamics**: Loss-gradients depend on $\nabla_f{\mathcal{L}}$, which fluctuates across checkpoints, necessitating multi-checkpoint tracking. Conversely, output-gradients remain stable under kernel behavior [7], which is described as NTK-like behavior in Sec. 3.1 in our manuscript, allowing a single backward pass to estimate a sample's global effect.
> > > 3. **Selection Strategy**: Due to stage-dependency, prior methods [1-3, 5] must track influence throughout training. In contrast, our output-gradient-based criterion serves as a stable, global proxy for training interactions without continual checkpoint monitoring. **This is exactly why our method differs conceptually from prior loss-gradient-based selection approaches.**
> > >
> > > ***Relation to LESS [5]*:**
> > >
> > > Importantly, LESS [5] discussed kernel behavior (App. F), However, it did not make NTK-style selection practical for LLMs due to the computational burden. **Our work pushes this line further** in two ways: (i) providing empirical evidence of stable NTK behavior in fine-tuned LLMs (Sec. 3.1), and (ii) making NTK-based selection efficient via Jacobian-free approximation, LoRA gradients, and random projection (Sec. 3.2-3.3).
> > >
> > > [1] GRAD-MATCH: Gradient Matching based Data Subset Selection for Efficient Deep Model Training, ICML 2021.
> > >
> > > [2] Gradient-Based Sample Selection for Black-Box Universal Domain Adaptation, AAAI 2025.
> > >
> > > [3] ClusterUCB: Efficient Gradient-Based Data Selection for Targeted Fine-Tuning of LLMs, EMNLP 2025.
> > >
> > > [4] Combatting dimensional collapse in llm pre-training data via diversified file selection, ICLR, 2025.
> > >
> > > [5] LESS: Selecting influential data for targeted instruction tuning, ICLR, 2024.
> > >
> > > [6] Data Shapley in One Training Run, ICLR Oral, 2025.
> > >
> > > [7] Neural Tangent Kernel: Convergence and Generalization in Neural Networks, NeurIPS, 2018.
> > >
> > > **We hope our clarifications and the additional analysis address your concerns. If you find our response satisfactory, we would be grateful if you could consider re-evaluating our submission and potentially raising the score.**

---

### Official Review · Reviewer_6Xiq · 2026-03-13

**Soundness:** 2
**Presentation:** 3
**Significance:** 3
**Originality:** 2
**Overall Recommendation:** 3
**Confidence:** 3

**Summary:**

This paper addresses the challenge of adapting LLMs to low-resource domains where task-specific data is scarce. The authors propose a new paradigm that mines useful auxiliary data from general-purpose corpora, such as Chain-of-Thought (CoT) collections, to improve domain performance. They introduce NTK-Selector, a two-stage framework grounded in Neural Tangent Kernel (NTK) theory. To make this practical for LLMs, they provide empirical evidence of "NTK-like" stability during fine-tuning and propose a Jacobian-free NTK approximation. Evaluations across medical, financial, legal, and psychological domains using Llama3-8B and Qwen3-8B show that NTK-Selector significantly outperforms domain-only fine-tuning and existing baselines like LESS and DSIR , achieving average gains of +8.7 points on Llama3-8B-Instruct.

**Compliance With Llm Reviewing Policy:**

Affirmed.

**Final Justification:**

Based on the rebuttal and other discussions, I tend to choose to keep the original score.

**Key Questions For Authors:**

1. Include a sensitivity analysis or a theoretical justification for the $10^{-5}$ scaling factor used in Equation 5 to ensure it is not a dataset-specific hyperparameter.
2. Provide a visualization (e.g., a T-SNE plot or a bar chart) showing the distribution of the original "topics" or "sources" in the CoT collection for the selected samples versus the rejected samples for at least two domains.
3. Conduct an ablation study where the domain data uses original "short" answers while the auxiliary data uses "CoT" answers to decouple the benefits of synthetic rationale generation from the benefits of NTK-based selection.

**Limitations:**

Acknowledge the computational overhead of the two-stage selection process and the reliance on a high-quality "warm-up" model as potential constraints for extremely resource-limited or noisy-data scenarios.

**Strengths And Weaknesses:**

**Strengths:**

1. The paper provides the first systematic empirical investigation into whether pre-trained LLMs exhibit NTK-like behavior during standard instruction tuning. This is significant because it moves NTK analysis from idealized, infinitely wide networks to real-world, finite-width models.

2. The method is validated on eight distinct tasks covering medicine, finance, law, and psychology, ensuring the findings aren't overfitted to a single domain. This demonstrates the high impact and generalizability of the proposed framework.

3. NTK-Selector shows consistent improvements even where other advanced baselines like LESS or TSDS fail or cause performance degradation. For example, on the RSDD task for Qwen3-8B, NTK-Selector is one of the few methods to show a positive gain.


**Weaknesses:**

1. The framework requires a "warm-up" LoRA training phase on the small domain dataset to compute embeddings for pre-selection. If the domain data is extremely noisy, this warm-up might bias the entire selection pipeline.

2. Table 13 shows that removing the warm-up stage results in a performance drop (76.6 to 76.3), yet the paper doesn't explore the sensitivity of the method to the hyperparameters of this warm-up.

3. The pre-selection size $M$ is shown to have a significant impact on accuracy, but the paper lacks a systematic way to determine the optimal $M$ without running the expensive second stage.

4. While the paper demonstrates that auxiliary CoT data is useful, there is no detailed analysis of what types of general data are being selected. For example, does the medical task select more "scientific" or "logical reasoning" samples from the CoT collection?

5. The reliance on GPT-4o-mini to generate CoT responses for the domain data introduces a potential "distillation" effect. It is unclear if the gains come from the NTK-Selector or the high-quality synthetic rationales.

6. The paper notes that random selection can sometimes degrade performance (e.g., Qwen3-8B on PrivacyQA), but it does not provide a deep error analysis of why certain "selected" samples might still be suboptimal.

---

> ### Author Rebuttal · Authors · 2026-03-31
>
> **We sincerely thank the reviewer for the insightful comments and the time. We respond point by point below.**
>
> > *W1: If the domain data is extremely noisy, this warm-up might bias the entire selection pipeline.*
>
> Re: **Noisy domain data is a real failure mode, but not unique to our method**: any target-conditioned selector can be affected if the target set itself is corrupted. Importantly, our method is only weakly dependent on warm-up: removing it changes the average result only from 76.6 to 76.3 (Table 13). Moreover, with or without warm-up, target and random OOD samples remain clearly separable in UMAP space [1], suggesting that warm-up does not distort NTK-based selection and that **NTK similarity may also help diagnose outliers/noisy target samples**.
>
> > *W2: The paper doesn't explore the sensitivity of the method to the hyperparameters of this warm-up.*
>
> Re: **Warm-up is used to build a more domain-aware embedding space for coarse pre-selection, not to maximize downstream performance itself**. Empirically it is not highly sensitive: removing it causes only a small drop (76.6→76.3, Table 13), while stronger domain-only training can overfit on some tasks (e.g., MedMCQA; Table 12).
>
> > *W3: The paper lacks a systematic way to determine the optimal pre-selection size without running the expensive second stage.*
>
> Re: We do not currently provide a closed-form rule, but **empirically the average NTK similarity of the second-stage selected subset is a useful proxy for the marginal value of increasing pre-selection size**. As shown in [2], when this score keeps increasing, downstream accuracy also tends to improve (e.g., MMLU-Med); when it saturates, downstream gains diminish (e.g., ContractNLI). In practice, this gives a cheap heuristic: enlarge the pre-selection pool and monitor the average second-stage NTK score, without fully retraining each time.
>
> > *W4: There is no detailed analysis of what types of general data are being selected.*
>
> Re: Appendix J already gives case studies showing that **NTK-Selector retrieves samples that are not same-distribution, but share similar question types and reasoning patterns.** We further analyze task-type distributions in [3], which show that NTK-Selector prefers reasoning types aligned with the target task.
>
> > *W5: It is unclear if the gains come from the NTK-Selector or the high-quality synthetic rationales.*
>
> Re: **Synthetic CoT is used for all methods in our comparisons**, including Domain-only and all baselines. Its role is simply to place domain and auxiliary data into a comparable reasoning-rich supervision format. Thus, NTK-Selector’s relative gains cannot be explained by rationale distillation alone, but rather stem from aligned reasoning patterns [3].
>
> > *W6: It does not provide a deep error analysis of why certain "selected" samples might still be suboptimal.*
>
> Re: Our current evidence suggests that **Random is unstable because its sampled auxiliary data is often poorly aligned with the target-domain learning signal**. As reflected in the same task-type analysis [3], random samples may be broadly “reasoning” data yet still differ substantially from the target in reasoning-type distribution, leading to unstable optimization directions that are sometimes helpful and sometimes harmful.
>
> > *Q1: Include a sensitivity analysis or a theoretical justification for the scaling factor to ensure it is not dataset-specific.*
>
> Re: **The scaling factor in Eq. (5) is used purely for numerical stability to avoid overflow during computation**. Since the same constant is applied to all samples, it only causes a global rescaling and therefore does not affect pairwise ranking or the final selection result.
>
> > *Q2: Provide a visualization showing the distribution of the original "topics" or "sources".*
>
> Re: We use Qwen3-32B to classify samples into 7 reasoning types and compare in [3]. We find: (i) domain data usually concentrates in one dominant category; (ii) Random reflects the global candidate-pool distribution and is similar across domains; (iii) NTK-Selector yields different distributions for different targets, indicating that its selections are systematically target-conditioned rather than accidental. One exception is depression diagnosis, where few Diagnosis-type candidates exist.
>
> > *Q3: Decouple the benefits of synthetic rationale generation from the benefits of NTK-based selection.*
>
> Re: Addressed in response to W5: since synthetic CoT is used for all methods, the relative gains of NTK-Selector cannot be explained simply by synthetic rationale distillation.
>
> **We are very enthusiastic about this research and would welcome the opportunity to discuss any remaining questions the reviewer might have.**
>
> [1] https://anonymous.4open.science/r/ICML2026_rebuttal-3883/UMAP_domain_ood/nowarmup.png
>
> [2] https://anonymous.4open.science/r/ICML2026_rebuttal-3883/NTK_sim_acc/ContractNLI.png
>
> [3] https://anonymous.4open.science/r/ICML2026_rebuttal-3883/selected_topic/LegalNLI.png

---

> > ### Author Rebuttal · Reviewer_6Xiq · 2026-04-04
> >
> > Thanks for your response. Based on other discussions, I tend to choose to keep the original score.

---

> > > ### Author Response · Authors · 2026-04-06
> > >
> > > We sincerely thank you for following our discussions. Since your last comment, there have been several significant developments in the discussion with other reviewers that we believe may address your hesitation:
> > >
> > > - **Growing Consensus:** Both Reviewer eWU8 and Reviewer gUWb have acknowledged our responses. Specifically, Reviewer ewU8 has explicitly stated that all the concerns are fully resolved and has committed to adjusting the score.
> > > - **Resolved Concerns:** Our latest detailed exchange with Reviewer HJcp has further solidified the conceptual foundation of our work. We successfully clarified that
> > >     1. **Optimization Paradigm:** The bi-level objective in Eq. (1) combined with a tractable proxy is the standard, rigorous formulation in data selection; our contribution lies in providing a superior proxy for low-resource settings.
> > >     2. **Output-Gradient vs. Loss-Gradient:** We have provided a granular analysis showing that while prior works (e.g., LESS) rely on loss-gradients (local/stage-dependent), our method operates with output-gradients (global/stable), capturing training dynamics more effectively.
> > >
> > > **Given that the "other discussions" you mentioned are now moving toward a positive consensus among several reviewers, we kindly ask you to reconsider our work. We believe the issues that caused your initial concerns have now been fully addressed across the panel.**
> > >
> > > Thank you again for your time and for helping us refine this work.

---

### Official Review · Reviewer_ewU8 · 2026-03-16

**Soundness:** 3
**Presentation:** 3
**Significance:** 3
**Originality:** 2
**Overall Recommendation:** 5
**Confidence:** 3

**Summary:**

This paper addresses the problem of adapting large language models (LLMs) to low-resource domains. The approach is mining useful auxiliary data from large-scale general-domain datasets (e.g. CoT 1.8M samples). The key observation is that this general chain-of-thought data shares reasoning patterns with domain tasks and can supplement scarce domain-specific data (1K samples). The authors propose a two-stage data selection framework grounded in the Neural Tangent Kernel (NTK); NTK-Selector. Stage 1 performs coarse-grained embedding-based pre-selection to reduce the candidate pool from millions to a manageable set. Stage 2 computes a Jacobian-free NTK approximation to estimate the alignment of training dynamics between candidate and domain samples. The NKT-Selector strategy demonstrates consistent performance gains one medical, financial, legal, and psychological datasets.

**Compliance With Llm Reviewing Policy:**

Affirmed.

**Final Justification:**

The addition of DiSF as a diversity-aware baseline helped the paper's core claim: DiSF performs near-random on several tasks, providing direct evidence that diversity-without-alignment is insufficient in this setting. Error bars have been provided and are within acceptable variance. The medical domain exclusion from eNTK regression is a legitimate methodological constraint. Given that most of my issues were addressed I'll increase my score given that these changes are made to the paper

**Key Questions For Authors:**

- The NTK utility score is defined as the average NTK similarity between a candidate and all domain points. This purely similarity-based criterion selects data points that are most aligned with the domain training set. How do the authors account for diversity within the selected subset?
- Have the authors measured the intra-set diversity of NTK-selected data compared to, e.g., DSIR or TSDS?
- Table 3 shows eNTK regression achieves >90% of fine-tuning accuracy on financial, legal, and psychological domains. Why is the medical domain excluded from this validation?

**Limitations:**

Yes

**Strengths And Weaknesses:**

### Strengths
1. **Well-Motivated Problem:** The paper addresses a genuinely important and practical problem: how to improve LLM performance in specialized domains where data is scarce, expensive, or privacy-constrained.
2. **Empirical Evaluation:** The experimental design is and well-controlled. All baselines share the same initialization, training procedure, and hyperparameters, differing only in selected data. Evaluating on 8 tasks across 4 domains with 2 model families (Llama3-8B-Instruct and Qwen3-8B) provides good evidence of generality
3. **Theoretical Foundation:** The NTK-based selection criterion has theoretical foundations. The formalization of NTK-like behaviour (Definition 3.1), the time-reparameterization theorem (Theorem 3.2), and the Jacobian-free NTK approximation (Definition 3.3) and demonstration (Fig. 2) provide meaningful insights

### Weaknesses
- **Missing baselines:** There are some recent baselines that would help strengthen the contribution of this research: **PDS** (Gu et al., ICLR 2025 Oral) which also has strong theoretical guarantees and demonstrated effectiveness on diverse LM tasks; **DiSF** (Fan et al., ICLR 2025) seeks to reduce the diversity-versus-similarity tradeoff in data selection using submodular optimization, which is directly relevant to NKT-Selectors greedy top-N selection that ignores diversity.
- **No Error Bars**: While the authors state results are "averaged over 3 runs," no standard deviations or confidence intervals are reported in Table 1.

---

> ### Author Rebuttal · Authors · 2026-03-31
>
> **We thank the insightful feed back from the reviewer and respond point by point as below.**
>
> > *W1: Missing baselines*
> >
>
> Re: Thank you for this suggestion. We respond in three parts.
>
> 1. **We added additional diversity-related baselines.** Following the reviewer’s suggestion, we additionally implemented DiSF (ICLR’25) on Llama3-8B-Instruct. Our original baselines also already included TSDS (NeurIPS’24), which partially accounts for diversity by downweighting near-duplicates.
> 2. **Our setting emphasizes target alignment rather than subset diversity alone.** In low-resource domain adaptation, the goal is to select auxiliary general-domain samples that are most useful for a specific target domain, rather than to construct a globally representative subset. Diversity is beneficial only if target relevance is maintained.
> 3. **The new results support this point.** TSDS is very close to plain embedding selection (74.8 vs. 74.9 on Llama3-8B-Instruct; 75.0 vs. 75.2 on Qwen3-8B), and the newly added DiSF results are consistently below NTK-Selector and in several cases close to Random. This suggests that improving subset diversity without explicitly modeling target-domain training alignment is insufficient in our setting.
>
> |  | MedMCQA | MMLU-Med | FPB | TFNS | Headline | ContractNLI | PrivacyQA | RSDD | Avg. |
> | --- | --- | --- | --- | --- | --- | --- | --- | --- | --- |
> | Llama3-8B-Instruct |  |  |  |  |  |  |  |  |  |
> | Random | 57.1 | 73.0 | 80.6 | 70.7 | 85.2 | 66.5 | 66.2 | 95.4 | 74.3 |
> | DiSF | 57.4 | 74.7 | 81.2 | 71.3 | 84.4 | 67.1 | 65.8 | 94.5 | 74.6 |
> | NTK-Selector | **59.1** | **73.8** | **85.5** | **73.3** | **86.2** | **70.6** | **67.8** | **96.5** | **76.6** |
> | Qwen3-8B |  |  |  |  |  |  |  |  |  |
> | Random | 60.8 | 82.6 | 65.2 | 71.9 | 80.9 | 77.3 | 53.9 | 90.8 | 72.9 |
> | DiSF | 61.2 | 82.1 | 77.8 | 70.2 | 81.0 | 78.0 | 56.1 | 93.3 | 74.6 |
> | NTK-Selector | **61.4** | **83.8** | **85.3** | **72.2** | **83.4** | **79.9** | **67.5** | **95.1** | 78.6 |
>
> For PDS, due to rebuttal-time constraints we could not complete a clean implementation in time, but we agree it is a valuable baseline and will include it in the revision.
>
> > *W2: No Error Bars*
> >
>
> Re: Thank you for pointing this out. Due to space limitations we reported only the means in Table 1. In [1] we provide the full mean±std results. We will include them in the appendix and clarify this in the revised version.
>
> [1] https://anonymous.4open.science/r/ICML2026_rebuttal-3883/mean_std/table.md
>
> >
> >
> >
> > *Q1: How do the authors account for diversity within the selected subset?*
> >
>
> Re: **We agree that diversity can matter, but its role depends on the data regime**. Many diversity-based methods aim to better cover the candidate pool, which is most natural when that pool is already matched to the target distribution. Here, however, the domain data is scarce and the large candidate pool is mainly OOD general data; there is no large same-distribution domain pool over which diversity can be meaningfully optimized. In this setting, increasing diversity may simply cover regions farther from the target domain like DiSF, so our method prioritizes target-aligned utility.
>
> > *Q2: Have the authors measured the intra-set diversity of NTK-selected data compared to, e.g., DSIR or TSDS?*
> >
>
> Re: **Yes. We sampled 1,000 selected examples from different methods and visualized them with t-SNE** [1]. Domain data forms relatively concentrated clusters, while NTK-selected auxiliary data is distributed around these clusters rather than collapsing to a single mode. This suggests variation around the target-domain manifold while remaining target-centered. The selected points do not lie entirely inside the domain clusters, which is expected because the auxiliary data comes from a general-domain corpus and is not strictly same-distribution.
>
> [1] https://anonymous.4open.science/r/ICML2026_rebuttal-3883/selected_diversity/selected_tsne.png
>
> >
> >
> >
> > *Q3: Table 3 shows eNTK regression achieves >90% of fine-tuning accuracy on financial, legal, and psychological domains. Why is the medical domain excluded from this validation?*
> >
>
> Re: **The omission is due to a limitation of the eNTK regression protocol, not selective reporting**. eNTK regression requires a label-aligned classification form with consistent label semantics across examples. This does not hold for MedMCQA and MMLU-Med, whose answer options vary across instances. Hence the Appendix F protocol is not directly applicable to these datasets. Medical is still included in all other major analyses, including NTK-like behavior (App. H.1), approximation error (App. H.2/H.3), multiple ablations (App. H.6/H.7/H.10). We will clarify this more explicitly in the revision.
>
> **Thank you again for the encouraging remarks regarding the importance of this problem. We believe the revised manuscript, incorporating your thoughtful suggestions, is much stronger. We remain open to any further feedback you may have.**

---

> > ### Author Rebuttal · Reviewer_ewU8 · 2026-04-05
> >
> > Thanks for engaging with my review. I'll adjust my score accordingly

---

> > > ### Author Response · Authors · 2026-04-06
> > >
> > > Thank you very much for your thoughtful follow-up and for indicating that our rebuttal has adequately addressed your concerns. We are very pleased to know that our responses satisfied your concerns and that you find the work deserving of a score adjustment.
> > >
> > > **We look forward to your updated evaluation. If there is anything else we can provide to support your final assessment, please do not hesitate to let us know.**

---

### Decision · Program_Chairs · 2026-04-30

**Decision:**

Accept (regular)

**Comment:**

The reviews reveal a meaningful split. Two reviewers were highly supportive, emphasizing that the paper addresses an important and practical problem, low-resource domain adaptation for LLMs, with a technically interesting and empirically strong solution. In their view, the NTK-based perspective is novel and well motivated, the approximations are carefully validated, and the gains across multiple domains, tasks, and model families are substantial. The rebuttal further strengthened this side of the case by adding diversity-related baselines, error bars, analyses of selected-data characteristics, and clarifications regarding theory, efficiency, and failure modes. These reviewers therefore saw the work as a rigorous and valuable contribution.

The two weaker reviews raised more foundational concerns, though of somewhat different kinds. One focused mainly on the gap between the formal bi-level objective and the practical proxy-based method, together with questions about novelty relative to prior gradient-based selection work. The other was more concerned with sensitivity to warm-up, pre-selection choices, possible confounds related to synthetic CoT data, and the need for additional analyses.  In my reading, the rebuttal addressed a substantial portion of these concerns - especially on baselines, efficiency, diversity, and empirical characterization - even if not every reviewer was fully convinced. Overall, this is clearly a borderline case, but it is also one with two genuine champions. When a paper has strong supporters who find both the problem and the contribution compelling, I tend to prefer giving it the opportunity to be presented rather than excluding it, particularly when the rebuttal is strong and the research problem is important and hard.